# Carbon nitride photocatalyzes regioselective aminium radical addition to the carbonyl bond and yields N-fused pyrroles

Bogdan Kurpil[1], Katharina Otte[1], Artem Mishchenko[2], Paolo Lamagni[3], Wojciech Lipiński[1], Nina Lock [3], Markus Antonietti[1] & Aleksandr Savateev [1]

Addition of N-centered radicals to C=C bonds or insertion into C–H bonds is well represented in the literature. These reactions have a tremendous significance, because they afford polyfunctionalized organic molecules. Despite the tetrahydroisoquinoline (THIQ) moiety widely occurring in natural biologically active compounds, N-unsubstituted THIQs as a source of N-centered radicals are not studied. Herein, we report a photocatalytic reaction between tetrahydroisoquinoline and chalcones that gives N-fused pyrroles—1,3-disubstituted-5,6-dihydropyrrolo[2,1-a]isoquinolines (DHPIQ). The mechanism includes at least two photocatalytic events in one pot: (1) C–N bond formation; (2) C–C bond formation. In this process potassium poly(heptazine imide) is used as a visible light active heterogeneous and recyclable photocatalyst. Fifteen N-fused pyrroles are reported with 65–90% isolated yield. DHPIQs are characterized by UV–vis and fluorescence spectroscopy, while the fluorescence quantum efficiency of fluorinated DHPIQs reaches 24%.

[1] Department of Colloid Chemistry, Max-Planck Institute of Colloids and Interfaces, Research Campus Golm, 14424 Potsdam, Germany. [2] V.I. Vernadsky Institute of General and Inorganic Chemistry, NAS of Ukraine, Palladina Avenue, 32/34, Kiev 03142, Ukraine. [3] Carbon Dioxide Activation Center, Interdisciplinary Nanoscience Center (iNANO), and Department of Chemistry, Aarhus University, DK-8000 Aarhus C, Denmark. Correspondence and requests for materials should be addressed to A.S. (email: oleksandr.savatieiev@mpikg.mpg.de)

Heterogeneous photocatalysis has emerged as a powerful technique empowering challenging chemical transformations[1]. The development of more efficient semiconducting photocatalysts includes, but is not limited to, band gap engineering, heterostructure design, and surface modification[2]. Surface properties of the heterogeneous photocatalyst are essential because they directly affect the interaction of the substrate with the surface of the photocatalyst and also influence the charge carriers mobility[3]. Thus, Chen and Hirao have shown that photocatalytic activity, for example, in hydroxylation of boronic acids to alcohols and oxidation of primary alcohols to aldehydes, may be boosted by adjusting the surface basicity of the photocatalyst[4,5].

It has been recently shown that carbon nitride heterogeneous photocatalysis is a useful instrument not only in water splitting[6,7], but also enables a variety of novel organic photoredox reactions that have been recently summarized in several reviews[8,9]. Potassium poly(heptazine imide) (K-PHI hereafter) is a member of the carbon nitride family with negatively charged nitrogen atoms[10,11]. This structural particularity has a great impact on the band structure and leads to a significantly more positive position (+2.54 eV vs. RHE) of the valence band[12]. Due to this feature, K-PHI shows high efficacy as the photocatalyst in different oxidation processes, i.e., the oxygen evolution reaction[13], oxidation of alcohols toward the synthesis of Hantzsch pyridines[14], synthesis of oxadiazoles[15] and thioamides[16], and oxidative thiolation of methylarenes[17].

Among a plethora of organic photoredox reactions, the addition of aminium radicals to C=C bonds or insertion into C–H bonds has a great importance as it serves the needs of synthetic organic chemistry and production of pharmaceuticals[18]. However, there are only few works on $R_2N^{\cdot}$ radicals addition to the polar C=O carbonyl bonds[19–21].

Synthesis and modification of tetrahydroisoquinoline (THIQ) derivatives have been attracting continually growing interest during the last decades. Photocatalytic single-electron oxidation of the amino group in THIQ, using homogeneous or heterogeneous visible light photocatalytic systems, leads to the formation of the aminium radical-cation[22] which is readily attacked at the α-carbon by different nucleophilic agents, for example nitromethane[23,24], cyanides[25], covalent enolates[26], alkynes[27], the Ruppert–Prakash reagents (TMSCF$_3$)[28,29], and even methylketones[30].

However, the scope of such photocatalytic transformations is limited solely to N-substituted THIQ and only allows functionalization of the C-2 atom of the substrate. On the other hand, THIQ with an unsubstituted NH-functionality has two reactive centers (C-2 α-carbon and N-1 nitrogen atom) and opens new possibilities to diverse types of reactions. Bifunctional compounds, such as chalcones, may be envisioned for use as reaction partners due to possessing a C=C double bond and a C=O carbonyl group in their structure[31,32]. In terms of photocatalysis, Yoon et al. have demonstrated that chalcones themselves, when activated by Lewis acids, can be used in cyclobutane synthesis[33–35]. On the other hand, Zeitler et al. suggested conditions mimicking enzymatic catalysis to activate enones toward the formation of cyclopentanes[36]. In these cases, however, the mechanism implies that the uncoupled electron is localized at the β-carbon atom, and therefore the reaction proceeds exclusively at this site.

Summarizing the literature, we envision that an appropriate photocatalyst might trigger a domino reaction between a chalcone molecule and THIQ. Such interaction may lead to the 1,3-disubstituted-5,6-dihydropyrrolo[2,1-a]isoquinolines (DHPIQs). DHPIQ derivatives occur in nature and are known as lamellarins[37]. Polyphenyl substituted DHPIQs were suggested for use as a hole transporting layer[38]. Preparation of such condensed heterocyclic compounds mainly demands microwave (MW) irradiation at high temperature[39] or using such exotic synthetic reagents as 3-phosphorylallenes[40] (Fig. 1).

Herein, we report the reaction between THIQ and chalcones mediated by the visible light active K-PHI photocatalyst without any additives (Fig. 1). This approach allows to alter the reactivity of the enone, that otherwise simply adds a THIQ via the classical aza-Michael reaction, and reveals the path for the synthesis of DHPIQs. Fluorinated DHPIQ derivatives exhibit strong fluorescence in the blue region, while the fluorescence internal quantum efficiency (IQE) reaches 24%.

## Results

**Optimization of the reaction conditions.** The photocatalyst, K-PHI, was synthesized according to the described procedure and its characteristics are given in Supplementary Figure 1[14]. Chalcones were synthesized according to the procedures given in Supplementary Methods. The photocatalytic coupling reaction between THIQ and chalcone **2a** was a starting point in our study, and all conditions and results of optimization experiments are presented in Table 1. In control reactions without the photocatalyst (entry 1) or without light irradiation (entry 2) no product was formed. Under light irradiation ($\lambda = 461$ nm, $51.7 \pm 0.03$ mW cm$^{-2}$, blue LED) at 80 °C with K-PHI as the photocatalyst after 20 h using 3 equivalents of THIQ and acetonitrile as solvent (entry 3), 87% of the starting chalcone **2a** was converted into a mixture of compounds: the desired dihydropyrrolo[2,1-a]isoquinoline (DHPIQ) **3a** and ketone **4a** (with ratio 1:1.1

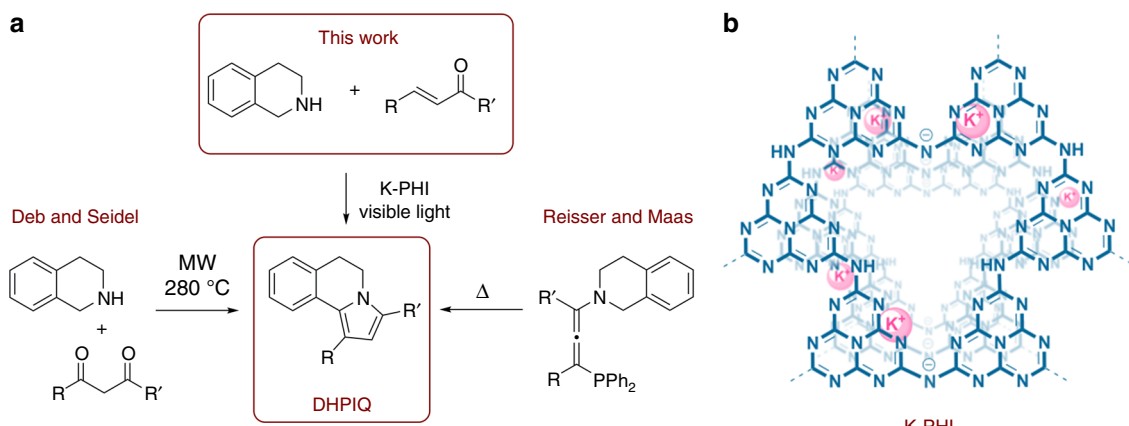

**Fig. 1** Key points of the work. Synthetic approaches to the DHPIQs (**a**) and chemical structure of K-PHI (**b**)

**Table 1 Optimization of the reaction conditions of the oxidative condensation between tetrahydroisoquinoline and chalcone 2a toward the synthesis of DHPIQ**

| Exp. no. | Solvent | T, °C | Time, h | Conv. of 2a, %[a] | 3a/4a[a] |
|---|---|---|---|---|---|
| 1[b] | CH₃CN | 80 | 20 | 0 | – |
| 2[c] | CH₃CN | 80 | 20 | 0 | – |
| 3[d] | CH₃CN | 80 | 20 | 87 | 1:1.1 |
| 4[e] | CH₃CN | 80 | 20 | 0 | – |
| 5 | CH₃CN | 80 | 20 | 93 | 8.2:1 |
| 6[f] | CH₃CN | 80 | 20 | 0 | – |
| 7[g] | CH₃CN | 80 | 20 | 0 | – |
| 8[h] | CH₃CN | 80 | 20 | 89 | 8:1 |
| 9[i] | CH₃CN | 80 | 20 | 84 | 8:1 |
| 10 | Dioxane | 80 | 20 | 69 | 2.2:1 |
| 11 | Benzene | 80 | 20 | 78 | 3.6:1 |
| 12 | t-BuOH | 80 | 20 | 62 | 2:1 |
| 13 | CH₃CN | 25 | 60 | 85 | 5.7:1 |
| 14[j] | CH₃CN | 80 | 20 | 91 | 8.2:1 |
| 15[k] | CH₃CN | 80 | 20 | 92 | 8.3:1 |
| 16[l] | CH₃CN | 80 | 20 | 91 | 8.2:1 |
| 17[m] | CH₃CN | 80 | 20 | 89 | 8.2:1 |
| 18[n] | CH₃CN | 80 | 20 | 0 | – |
| 19[o] | CH₃CN | 80 | 20 | 35 | 4.3:1 |
| 20[p] | CH₃CN | 80 | 20 | 85 | 13:1 |

Reaction conditions: chalcone (50 μmol), THIQ (3 eq., 150 μmol), K-PHI (5 mg), acetone (3 eq., 9 mg) solvent (2 mL), λ = 461 nm, argon atmosphere
(a) The conversion of **2a** and the molar ratio between **3a** and **4a** was determined by GC-MS; (b) without photocatalyst and acetone; (c) without light irradiation and acetone; (d) without acetone; (e) reaction in atmosphere of oxygen; (f) without photocatalyst; (g) without light irradiation; (h) 10 mg and (i) 2 mg of the photocatalyst; (j) second run; (k) third run; (l) fourth run; (m) Ir(ppy)₃ (2 mg, 3 μmol); (n) Ru(bpy)₃Cl₂·6H₂O (2.2 mg, 3 μmol); (o) mpg-CN (5 mg) instead of K-PHI; (p) Na-PHI (5 mg) instead of K-PHI

correspondingly). This result could be explained by the stoichiometry of this transformation—the formation of compound **3a** demands interaction of THIQ and chalcone **2a** with the abstraction of $H_2O$ and $H_2$ molecules. The starting chalcone **2a** is then reduced by in situ released hydrogen to the corresponding ketone **4a**. In other words, the chalcone **2a** plays the role of electron/proton scavenger under the photocatalytic conditions (see also the suggested mechanism below). When **1a**-$d_2$ (98% of $d$-labeled compound) was used as a reagent under the identical photocatalytic conditions, along with DHPIQ **3a** we have detected a mixture of $d$-labeled ketones **4a**-$d_1$ by mass-spectrometry (Supplementary Figure 2), suggesting that indeed under the photocatalytic conditions THIQ acts as a reductant. Using oxygen as electron scavenger (entry 4), no DHPIQ **3a** was obtained. However, THIQ was completely oxidized to 3,4-dihydroisoquinoline (DHIQ), while the chalcone **2a** remained intact. Such observation might be explained comparing standard redox potential of the couple $O_2/O_2^{·-}$ (−0.57 V vs. saturated calomel electrode (SCE))[41] and reduction potential of chalcones ($E_{red} =$ −1.0…−1.5 V vs. SCE). Reduction of oxygen is a more favorable process compared to the reduction of chalcones. Therefore, in the presence of $O_2$, reduction of chalcones is completely suppressed. The generated $O_2^{·-}$ is reactive enough to oxidize THIQ to DHIQ[42]. Following these preliminary experiments, it was decided to use acetone as electron scavenger (entry 5). Under these mild conditions, high conversion (93%) of chalcone **2a** and high selectivity (89%) with respect to DHPIQ **3a** were achieved (see the

mechanism discussion below for more details about the role of acetone). Using both smaller and larger amounts of the photocatalyst (entries 8–9) gave slightly lower conversions (84–89%) of chalcone, but almost the same selectivity toward DHPIQ **3a**. In less polar solvents such as dioxane, benzene, and tert-butanol (entries 10–12), the conversion and selectivity were lower, possibly due to poor stabilization of the polarized transition state structures. This photocatalytic reaction could also be performed at room temperature (entry 13), but it takes more time (60 h) to get conversion (85%) and selectivity (85%) comparable to the experiments at 80 °C. After recycling K-PHI for three times (entries 14–16), the conversion of the chalcone has slightly decreased (91%) nevertheless the selectivity was the same (89%). Ir(ppy)₃ gave DHPIQ with 89% selectivity (entry 17), but no product was obtained when Ru(bpy)₃Cl₂·6H₂O was employed as a photoredox catalyst (entry 18) presumably due to a lower reduction potential of the excited state. In case of mesoporous graphitic carbon nitride (mpg-CN, entry 19), 35% conversion of chalcone **2a** was obtained. Another representative of poly(heptazine imide)s, Na-PHI, gave 85% conversion, but higher selectivity (entry 20). The difference in performance of these two materials might be explained by variation in their microstructure as well as surface properties[43].

**Expanding the scope of the substrates**. The effectiveness of the photocatalytic approach was proved for the wide range of

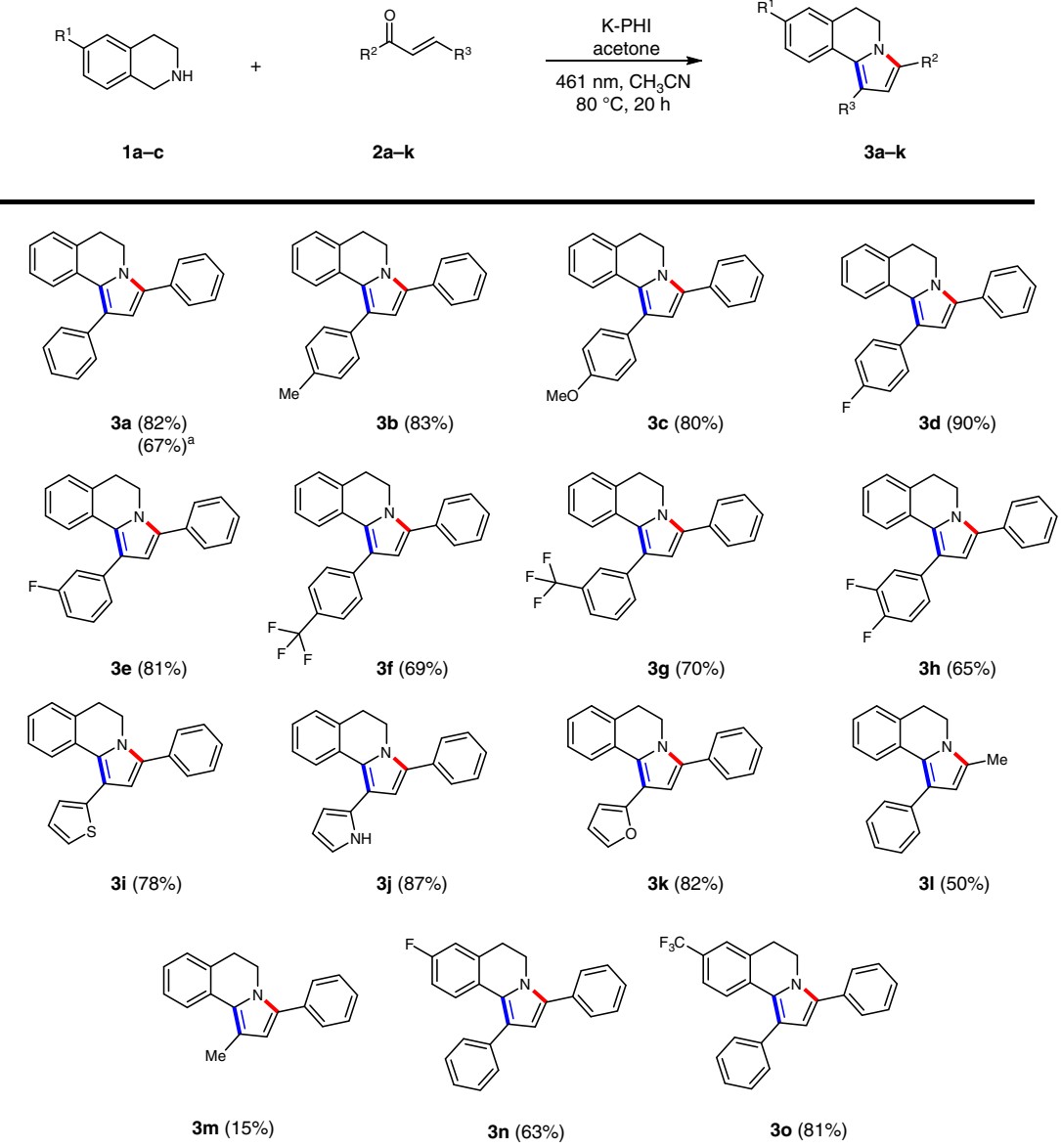

**Fig. 2** Scope of dihydropyrrolo[2,1-a]isoquinolines **3a–o** synthesized via a photocatalytic cyclocondensation tandem reaction. Isolated yields are given in brackets. Superscript "a" indicates the reaction was scaled using 300 mg of the chalcone **2a**

chalcones **2a–m** (Fig. 2). Aryl- and hetaryl-substituted chalcones **2a–k** gave the respective DHPIQs from good to excellent yields (65–90%). In case of chalcones **2f–h** the significant formation of ketones **4f–h** as side products was detected, which lowered the yield of **3f–h** to 65–70%. These results may be explained by the enhanced electron scavenging ability of the chalcones **2f–h**, bearing electron withdrawing (trifluoromethyl)phenyl or 3,4-di-fluorophenyl substituents, which make them competitive to acet-one in the electron capture and further reduction process. Chal-cones bearing at least one methyl group, **2l** and **2m**, also gave the corresponding DHPIQs, **3l** and **3m**, although with lower yields. On the other hand, (E)-hex-4-en-3-one did not react under similar conditions, partially due to its more negative reduction potential. (E)-4,4-dimethyl-1-phenylpent-2-en-1-one also did not yield the corresponding DHPIQ, presumably due to steric hindrance of the *tert*-butyl group that obstructs coupling of the chalcone with THIQ. Finally, we have also explored the possibility to use sub-stituted THIQ. Thus, 6-fluoro- and 6-trifluoromethyl-THIQ gave the corresponding DHPIQs, **3n,o**, with 63% and 81% yield, while electron rich 6,7-dimethoxy-THIQ was completely oxidized to the

**Fig. 3** Synthesis of reduced DHPIQs

dihydroderivative, but no DHPIQ was formed. Among other amines, only dibenzylamine and *N*-benzylbutan-1-amine activated by a phenyl ring, but having rather flexible structure compared to THIQ **1a**, gave the respective products of condensation, although in trace amounts of 15% and 3%, respectively.

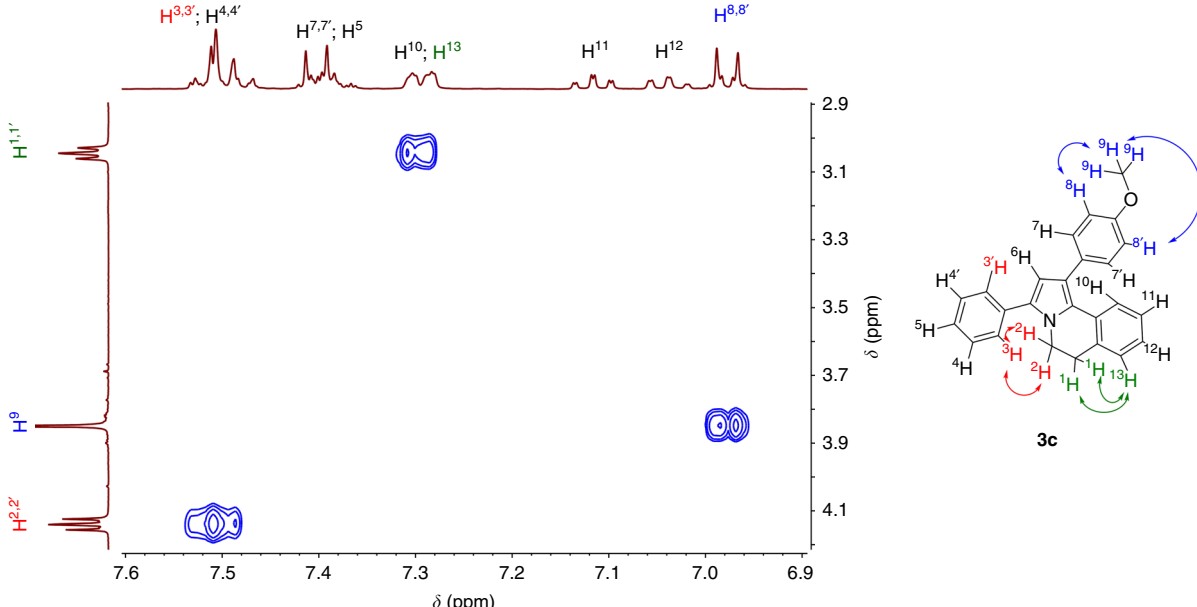

**Fig. 4** Fragment of the NOESY spectrum of DHPIQ **3c**

Different reactivity was observed for chalcone **2p** (Fig. 3). Instead of DHPIQ, the saturated analogue **7p** was isolated, though in low yield. When **1a**-$d_2$ was employed, the corresponding compound **7p**-$d_1$ was obtained. Compounds **7p** apparently are stabilized by hydrogen bonding, which obstructs dehydration and dehydrogenation. This experiment gave us an idea about the possible mechanism of DHPIQ formation.

In the NOESY spectrum of DHPIQ **3c**, a through-space interaction between the *ortho*-phenyl protons and the methylene group of the THIQ core is observed supporting the structural correctness (Fig. 4).

**Basic chemical and optical properties of the DHPIQs.** Furthermore, we have shown that DHPIQs, as exemplified by compound **3d**, smoothly undergo electrophilic substitution using N-bromosuccine imide (NBS) as a reagent at room temperature (Fig. 5). The bromo-derivative **8d** may be envisioned as a useful partner for cross-coupling reactions.

DHPIQs synthesized here are colorless substances and absorb light below 360–370 nm (Fig. 6a).

These compounds exhibit pronounced fluorescence in the blue region when excited with UV light (Fig. 6b, c). The emission maximum depends on the structure of DHPIQ and is in the range 399–431 nm (Table 2). Notably, CF$_3$-substituted DHPIQ **3f** that fluoresces at 431 nm also has an IQE of fluorescence of 24%.

**Redox properties of the chalcones and THIQ.** In order to determine the redox properties of the reagents and therefore shed light on the photocatalytic mechanism, we have performed a cyclic voltammetry (CV) study (Fig. 7). Thus, THIQ **1a** is stable on the reduction side at least up to −1.8 V vs. SCE, however, typically for aliphatic amines, oxidation starts at ca. +0.8 V vs. SCE. The similar oxidation potential was detected for 6,7-dimethoxy-1,2,3,4-tetrahydroisoquinoline. On the other hand, chalcones **2a–d** and **2i** showed stability against reduction up to −1.2 V vs. SCE and the tendency is: the more electron deficient the chalcone the less negative reduction potential it has. In agreement with this, chalcone **2q**, derived from 2,3,4,5,6-penta-fluorobenzaldehyde, has a reduction potential of −1.0 V vs. SCE, reflecting high affinity to electrons. Under the photocatalytic

**Fig. 5** Derivatization of DHPIQs by electrophilic bromination with NBS

conditions **2q** was quantitatively reduced to ketone **4q** via a proton-coupled electron transfer (PCET) and did not give the corresponding DHPIQ. Methyl-substituted chalcones **2l** and **2m** have more negative reduction potentials −1.45 and −1.50 V, respectively. The yield of DHPIQ **3l** was still higher than **3m**, indicating the importance of an aromatic substituent in position 3 of the chalcone for efficient reduction to the radical anion. *Tert*-butyl-substituted chalcone showed a reversible reduction peak at −1.67 V, implying that the subsequent reaction, e.g., coupling of the radicals, is inhibited and at the given scan rate of 50 mV s$^{-1}$ the radical anion is oxidized back to the chalcone. Finally, (*E*)-hex-4-en-3-one has a reduction potential below −1.75 V vs. SCE. The reduction power of K-PHI is apparently not sufficient to reduce aliphatic chalcones. Therefore no DHPIQ was obtained in this case.

In addition, chalcones are stable against oxidation up to at least +1.5 V vs. SCE. Only chalcone **2c** showed an irreversible oxidation peak at this potential. Summarizing the CV study, under the photocatalytic conditions it is more likely that photogenerated holes (+1.96 V vs. SCE) oxidize THIQ **1** ($E_{ox} \leq$ +0.8 V vs. SCE) rather than any of the chalcones ($E_{ox} >$ +1.5 V vs. SCE). In order to comply with the balance of electrons, the photogenerated electrons are taken by the chalcones ($E_{red} \geq$ −1.5 V vs. SCE) rather than THIQ ($E_{red} \leq$ −1.8 V vs. SCE).

**DFT calculations.** Absolute energies of DHPIQ and chalcone frontier orbitals were calculated using the PBE0-D3 functional and are shown in Fig. 8a. Although they do not correspond to the real redox potentials of these compounds obtained from CV in acetonitrile, the tendency is the same, i.e., THIQ is more

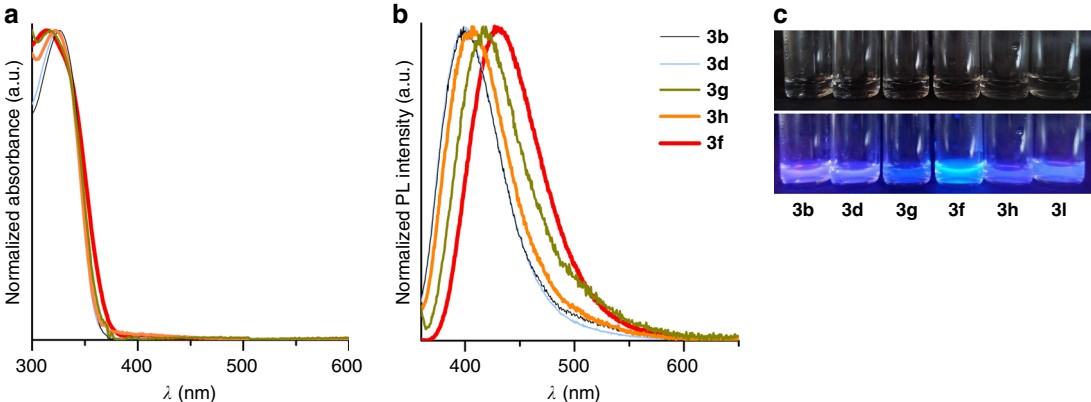

**Fig. 6** Basic optical properties of the DHPIQs. **a** UV–vis absorption spectra of DHPIQ solution in MeCN (ca. $10^{-4}$ M). **b** Fluorescence spectra of DHPIQ solution in MeCN (ca. $10^{-4}$ M) were obtained upon sample excitation with 350 nm. **c** Appearance of DHPIQ solutions under ambient light and under UV (365 nm)

| Table 2 Spectroscopic data of DHPIQs | | | |
|---|---|---|---|
| **DHPIQ** | $\lambda_{abs}$, nm | $\lambda_{em}$, nm | IQE, % |
| **3b** | 326 | 399 | 17 |
| **3d** | 324 | 399 | 21 |
| **3f** | 316 | 431 | 24 |
| **3h** | 322 | 407 | 16 |
| **3g** | 314 | 419 | 13 |
| The IQE was measured using an integrating sphere upon sample excitation with 360 nm | | | |

susceptible for oxidation, while the chalcone is more likely to work as electron acceptor. The HOMO of THIQ is localized largely on the saturated part of the molecule, while the aromatic ring contributes mostly to the LUMO. The LUMO orbital of the chalcone is delocalized over the whole molecule and this might explain why chalcone **2a** relatively easily undergoes reduction. Similar results were obtained with the B3LYP-D3 and CAM-B3LYP functionals (Supplementary Tables 1–4).

Figure 8b represents the distribution of atomic charges and spin population in THIQ **1a**, chalcone **2a**, and their charged radicals. Thus, in $[\mathbf{1}]^{\cdot+}$ the unpaired electron is localized mainly on the $N$-atom (0.61e) (Supplementary Figure 3, Supplementary Table 1). The acidity of the N–H proton in $[\mathbf{1}]^{\cdot+}$ is higher compared with the neutral THIQ, as the partial positive charge at the H-atom increased on average from +0.25 to +0.35 depending on the functional (Supplementary Table 2). Bearing this in mind, the structure of the $N$-centered aminium radical cation may be proposed as an intermediate in photocatalytic oxidation of THIQ.

In case of $[\mathbf{2a}]^{\cdot-}$, the uncoupled electron is delocalized over the carbonyl O- and C-atoms and the C-atom in β-position to the carbonyl group, but not located at the C-atom in α-position to the carbonyl group (Supplementary Table 3). Therefore, only resonance structures $[\mathbf{2a'}]$, $[\mathbf{2a''}]$, and $[\mathbf{2a'''}]$ of the chalcone radical anion may be proposed.

## Discussion

Taking into account the structures of THIQ, chalcone and the respective DHPIQ, the reaction requires the following steps: (1) C–N bond formation between THIQ and the carbon atom of the C=O group; (2) C–C bond formation between THIQ and the chalcone; (3) elimination of a H$_2$ molecule; and (4) elimination of a H$_2$O molecule. Based on the experimental data as well as DFT calculations discussed above, Fig. 9 outlines the possible sequence

of the reactions that leads to the formation of DHPIQ from THIQ and chalcone. A detailed scheme of the proposed mechanism is given in Supplementary Figure 4.

Upon excitation with light, K-PHI is converted into the excited K-PHI* species that eventually follows the path of reductive quenching. One-electron oxidation of THIQ **1** gives the corresponding radical cation $[\mathbf{1}]^{\cdot+}$, a common intermediate in organic redox chemistry[44], and the long-lived K-PHI$^{\cdot-}$ radical anion that was detected by EPR in situ (Supplementary Figure 5)[45].

Single electron transfer from K-PHI$^{\cdot-}$ to the chalcone gives the radical anion $[\mathbf{2}]^{\cdot-}$. Coupling of the radicals $[\mathbf{1}]^{\cdot+}$ and $[\mathbf{2}]^{\cdot-}$ accompanied by the photocatalytic oxidation of the intermediate **5** leads to the radical cation $[\mathbf{5}]^{\cdot+}$. Although hemiaminals are not stable compounds, Li et al. proposed them as intermediates in the synthesis of arylamines from cyclic ketones and amines[46]. In the control experiment without photocatalyst and/or light, THIQ and chalcones did not produce **5**, but formed the aza-Michael adduct **9** (Supplementary Figures 6,7, Supplementary Notes 1,2). The aza-Michael adduct **9d** under the optimized photocatalytic conditions did not yield the corresponding DHPIQ (Supplementary Figure 8, Supplementary Note 3), implying that **9d** is not the intermediate of DHPIQ **3d** formation.

PCET from $[\mathbf{5}]^{\cdot+}$ to acetone gives the iminium cation **6** that via a Mannich-like cyclization and subsequent hydride transfer is converted to intermediate **7**. On the last step, DHPIQ is obtained from **7** by sequential dehydrogenation and dehydration. Evolution of hydrogen was detected by GC-TCD (Supplementary Figure 9).

In order to check whether DHPIQ **3a** can be synthesized from other building blocks and presumably get more insight into the mechanism, we tried the following combinations of precursors: (1) 3,4-dihydroisoquinoline and chalcone **2a**; (2) 3,4-dihydroisoquinoline and 1,3-diphenylprop-2-yn-1-one; (3) acetophenone, benzaldehyde and THIQ; (4) styrene, benzaldehyde and THIQ (Supplementary Notes 4–8). In all these cases, no DHPIQ was formed indirectly supporting the proposed scheme of the mechanism.

In the present account, we have shown that K-PHI, a transition metal-free, visible light responsive, and heterogeneous carbon nitride material, triggers a tandem photocatalytic reaction between THIQ and chalcones. Fifteen DHPIQs were isolated with 15–90% yield. The reaction reported herein gives the access to polycyclic molecules that exhibit strong fluorescence in the blue region with IQE up to 24% and might be used, for example, in organic photovoltaics. The DHPIQ core can be subsequently functionalized using electrophilic substitution reactions.

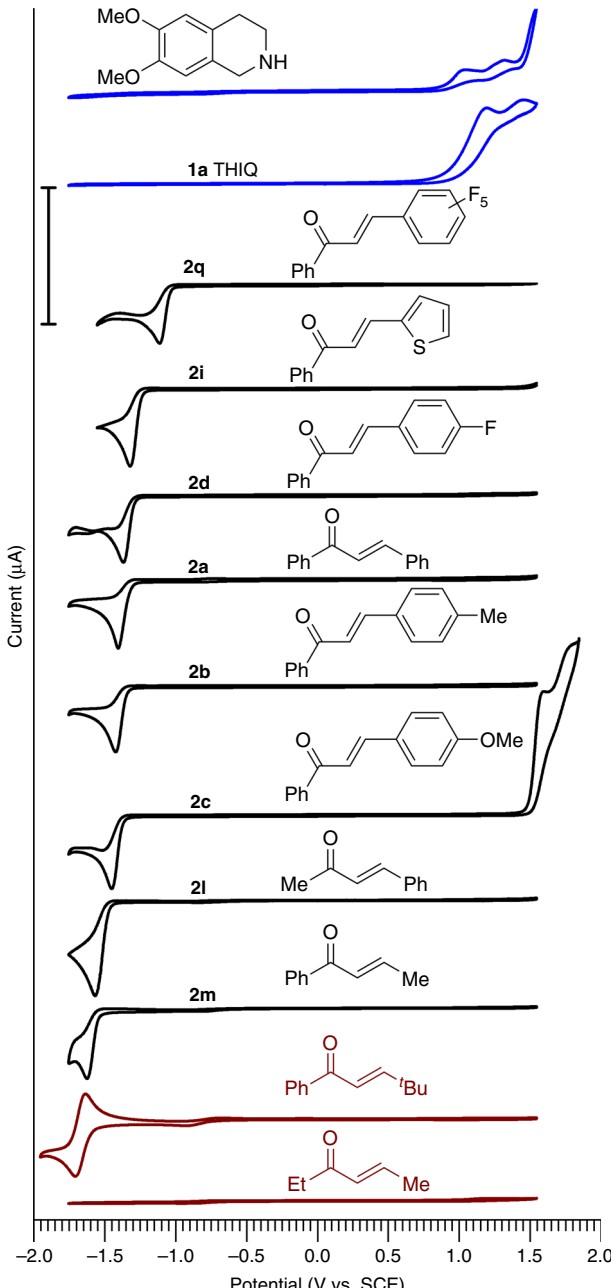

**Fig. 7** Cyclic voltammograms of THIQ **1a**, 6,7-dimethoxy-1,2,3,4-tetrahydroisoquinoline, chalcones **2a–d,i,l,m,q** recorded in MeCN at the scan rate 50 mV s⁻¹ at room temperature (20–22 °C). Tetrabutylammonium tetrafluoroborate in MeCN (0.1 M) was used as a supporting electrolyte. Scale bar corresponds to 10 μA

## Methods

**Compounds characterization.** ¹H and ¹³C NMR spectra were recorded on Agilent 400 MHz (at 400 MHz for Protons and 101 MHz for Carbon-13). Chemical shifts are reported in ppm vs. solvent residual peak: chloroform-*d* 7.26 ppm (¹H NMR), 77.1 ppm (¹³C NMR); acetonitrile-*d₃* 1.94 ppm (¹H NMR), 118.3 ppm (¹³C NMR). High-resolution mass spectral data were obtained using Waters XEVO G2-XS QTOF with Acquity H-Class (HPLC). UV–vis absorption spectra of DHPIQ in MeCN (ca. 10⁻⁴ M) were recorded using a T-70 spectrometer. Steady-state fluorescence spectra of DHPIQ in MeCN (ca. 10⁻⁴ M) were recorded using a Jasco FP-8300 fluorescence spectrometer equipped with integrating sphere. The samples were excited with 350 nm. Fluorescence IQE was measured using the integrating sphere upon samples excitation with 360 nm.

**Photocatalyst characterization.** Fourier transform infrared (FT-IR) spectra were recorded on Thermo Scientific Nicolet iD5 spectrometer. Powder X-ray diffraction

patterns were measured on a Bruker D8 Advance diffractometer equipped with a scintillation counter detector with CuKα radiation (λ = 0.15406 nm) applying 2θ step size of 0.05° and counting time of 3 s per step. Nitrogen adsorption/desorption measurements were performed after degassing the samples at 150 °C for 20 h using a Quantachrome Quadrasorb SI-MP porosimeter at 77.4 K. The specific surface areas were calculated by applying the Brunauer–Emmett–Teller (BET) model to adsorption isotherms for $0.05 < p/p_0 < 0.3$ using the QuadraWin 5.11 software package. Scanning electron microscopy (SEM) images were obtained on a LEO 1550-Gemini microscope. The X-ray photoelectron spectroscopy (XPS) measurements were carried out in an ultrahigh vacuum (UHV) spectrometer equipped with a VSW Class WA hemispherical electron analyzer. A dual anode Al Kα X-ray source (1486.6 eV) was used as incident radiation. Survey and high-resolution spectra were recorded in constant pass energy mode (44 and 22 eV, respectively). During the UPS (He I excitation energy $h_\nu$ = 21.23 eV) measurements a bias of 15.32 V was applied to the sample, in order to avoid interference of the spectrometer threshold in the UP spectra. The values of the valence band maximum (VBM) are determined by fitting a straight line into the leading edge. Optical absorbance spectra of powders were measured on a Shimadzu UV 2600 equipped with an integrating sphere. Emission spectra were recorded on Jasco FP-8300 instrument. The excitation wavelength was 360 nm. The TEM measurements were acquired using a double-corrected Jeol ARM200F, equipped with a cold field emission gun and a Gatan GIF Quantum. The used acceleration voltage was 200 kV and the emission was set to 10 μA in order to reduce beam damage. An objective aperture with a diameter of 60 μm was introduced into the beam to improve the contrast while still allowing for atomic resolution.

**Irradiance of the LED modules.** Irradiance was measured using PM400 Optical Power and Energy Meter equipped with the integrating sphere S142C.

**Hydrogen (H₂) detection.** The experiment was performed using Agilent Technologies 7890B gas chromatography system equipped with a thermal conductivity detector (TCD) allowed for analysis of the gas produced during the photocatalytic experiment. The separation of the gaseous species was performed with an Agilent select permanent gases/CO₂ capillary column set. The latter was consisting of two parallel columns that combine CP Molsive 5 Å for permanent gas analysis, and CP PoraBOnD Q for CO₂ analysis. The detector and oven temperatures used were 200 and 45 °C, respectively. Ar was used as a carrier phase with a flow rate of 14 mL min⁻¹. The injection was performed with a 250-μL gas-tight syringe from SGE Analytical Science. A calibration gas consisting of 20% CO₂, 5% CO, 5% CH₄, 2% H₂, and 1% C₂H₆ mixed in Ar, injected in known volumes, was used to obtain the calibration curve that allowed the quantification of the gas products.

**Electrochemistry.** CV measurements were performed in a glass single-compartment electrochemical cell. A glassy carbon disc electrode (1 mm in diameter) was used as working electrode, and a Pt wire as counter electrode. The experiments were performed using a Ag/AgCl reference electrode, and the potential values were then converted to the SCE reference system according to the equation $E_{SCE} = E_{Ag/AgCl} - 0.045$ V. Each compound was studied in a 2 mM concentration in a 0.1 M tetrabutylammonium tetrafluoroborate (TBABF₄)/acetonitrile electrolyte solution (10 mL). Before voltammograms were recorded, the solution was saturated with Ar, and an Ar flow was kept in the headspace volume of the electrochemical cell during CV measurements. A potential scan rate of 0.050 V s⁻¹ was chosen, and the potential window ranging from +1.5 $V_{SCE}$ to −1.8 $V_{SCE}$ (and backwards) was investigated. CV was performed under room-temperature conditions (~20–22 °C).

**Synthesis of K-PHI.** The procedure is similar to the method reported earlier with minor changes[14,15]. A blend of potassium chloride (4.54 g), lithium chloride (3.71 g), and 5-aminotetrazole (1.65 g) was grinded in a ball mill using the following parameters—shaking rate 25 Hz, time 5 min. The flour-like powder was heated in a porcelain crucible covered with a lid under nitrogen flow (15 L min⁻¹). The temperature program was the following: heating from room temperature to 550 °C for 4 h, calcination at 550 °C for 4 h. The crucibles were spontaneously cooled to room temperature. The cake and deionized water (100 mL) were brought together in a beaker and stirred at room temperature for 3 h. Solid was filtered, thoroughly washed with water and dried in vacuum (20 mbar) at 50 °C for 15 h.

**A general method of DHPIQs (3a–o) preparation.** A glass tube with rubber-lined cap was evacuated and filled with argon three times. To this tube, THIQ (20 mg, 150 μmol), corresponding chalcone (50 μmol), acetone (9 mg, 150 μmol), and K-PHI (5 mg), and acetonitrile (2 mL) were added. The resulting mixture was stirred at 80 °C under irradiation of blue LED (λ = 461 nm, 51.7 ± 0.03 mW cm⁻²) for 20 h. Then the reaction mixture was cooled to room temperature and centrifuged, a clear solution was separated and the solid residue was washed with acetonitrile (2 mL) and centrifuged again. The organic solutions were combined and evaporated to dryness. The residue after evaporation was purified by silica gel column chromatography using a mixture of hexane/diethyl ether (98:2) as an eluent.

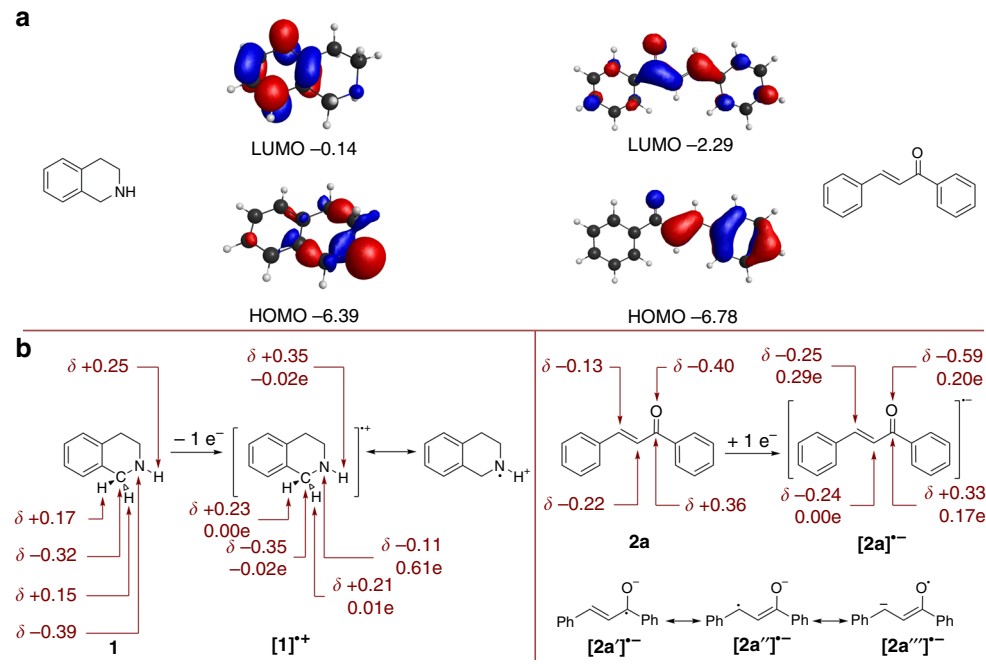

**Fig. 8** The results of DFT PBE0-D3 calculations. **a** Absolute energies of the frontier molecular orbitals (in eV vs. vacuum level) of THIQ **1a** and chalcone **2a** and their spatial representation. **b** Distribution of atomic charges ($\delta$) in neutral molecules, THIQ radical cation [**1a**]$^{\bullet+}$ and chalcone radical anion [**2a**]$^{\bullet-}$. Mulliken spin population (in fraction of elementary charge) in THIQ radical cation [**1a**]$^{\bullet+}$ and chalcone **2a** radical anion [**2a**]$^{\bullet-}$

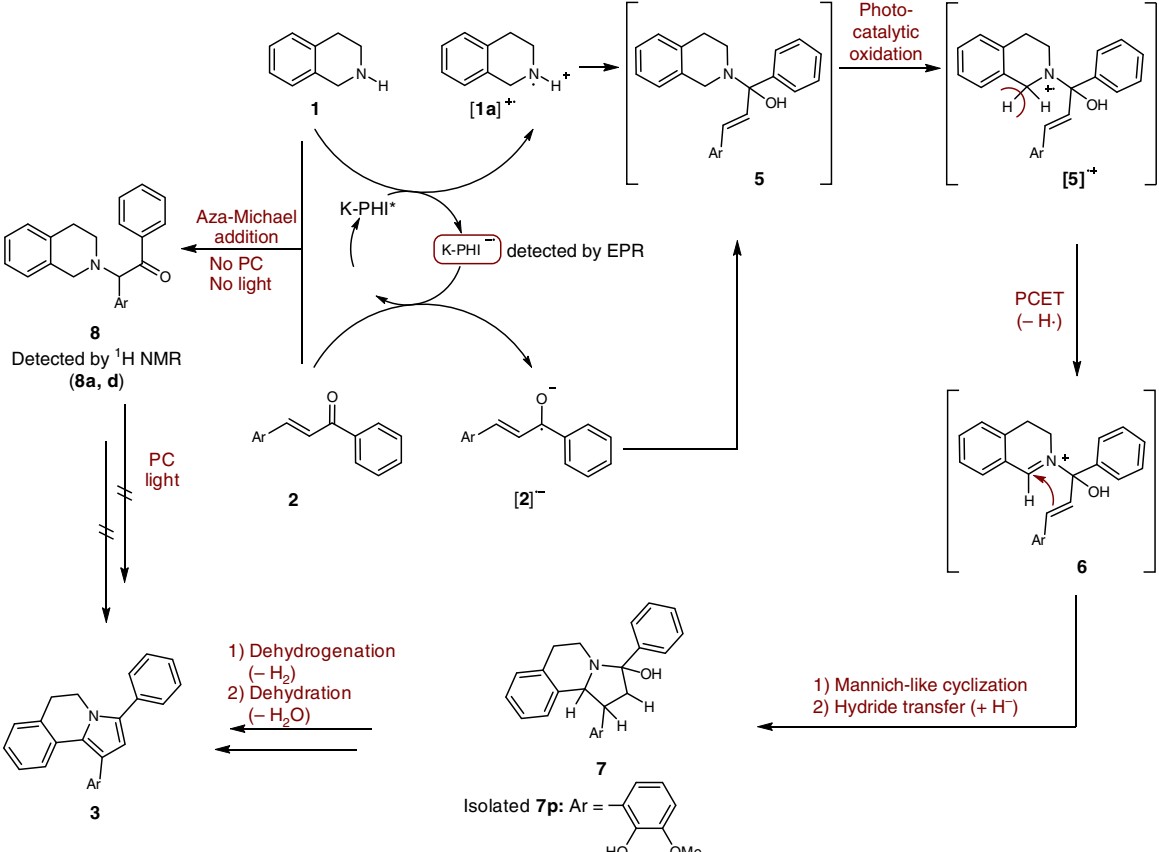

**Fig. 9** A proposed sequence of chemical reactions—photocatalytic synthesis of DHPIQ. PC stands for photocatalyst

**Computation details**. The DFT calculations for neutral molecules [**1**], [**2a**] and corresponding radical ions [**1**]$^{\cdot+}$, [**2a**]$^{\cdot-}$ were performed using three hybrid functionals: PBE0, B3LYP (both with D3 dispersion correction), and CAMB3LYP. The basis was Dunning–Hay double zeta set with one $d$-type polarization function on nonhydrogen atoms and one $p$-type function on hydrogen. For nitrogen and oxygen, the basis set was additionally extended by a diffuse $sp$-shell. Unrestricted Kohn–Sham approach was used for [**1**]$^{\cdot+}$ and [**2a**]$^{\cdot-}$ to find the energy minima of their ground doublet states. Geometry optimization for all species was performed with the gradient convergence threshold of $1 \cdot 10^{-6}$ Hartree/Bohr and the local energy minimum character of the optimized structures was confirmed by Hessian calculation. All calculations were carried out in the GAMESS (US) program package[47,48].

## Data availability

The data that support the findings of this study are available from the corresponding author upon reasonable request.

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

## Acknowledgements

The authors are grateful for the financial support from the Deutsche Forschungsgemeinschaft (DFG-An 156 13-1) and the Danish National Research Foundation (Carbon Dioxide Activation Center, DNRF 118). A.M. acknowledges the financial support from the scholarship of the President of Ukraine for young scientists. Computing resources for DFT calculations were provided by the SCIT supercomputer (V.M. Glushkov Institute of Cybernetics of the NAS of Ukraine)[49]. The authors thank Dr. C. Teutloff for acquiring EPR spectra.

## Author contributions

B.K. conceived the idea, synthesized chalcones, performed photocatalytic experiments, purified and characterized DHPIQs, prepared a draft of the manuscript. K.O. performed photocatalytic experiments. A.M. performed DFT calculations. P.L. synthesized K-PHI photocatalyst, performed CV study, performed analysis of the gas headspace after the photocatalytic experiments using GC-TCD. W.L. scaled up the photocatalytic experiment, did experiments to investigate the mechanism, acquired UV–vis, PL spectra of DHPIQ in acetonitrile and IQE measurements. N.L. and M.A. contributed to the manuscript preparation. A.S. performed bromination of DHPIQ with NBS, suggested the mechanism of the photocatalytic reaction, finalized and submitted the manuscript and supplementary information.

## Additional information

**Competing interests:** The authors declare no competing interests.

