## [Peer Review File · Nature Communications]

Editorial Note: Parts of this peer review file have been redacted as indicated to remove third-party material where no permission to publish could be obtained.

Reviewers' comments:

Reviewer #1 (Remarks to the Author):

The manuscript from Antonietti, Savateev and coworkers reported the potassium poly(heptazine imide) photocatalyzed aminium radical addition to the carbonyl bond for the synthesis of 1,3-disubstituted-5,6-dihydropyrrolo[2,1-a]isoquinolines (DHPIQ). Most N-centered radicals are derived from α -Amido-oxy, azirine, sulfonamide, hydrazine or hydrazone. There are very few reports about N-unsubstituted tetrahydroisoquinolines as a source of N-centered radicals. In this manuscript the authors show that NH-THIQ are viable coupling precursors with chalcones, generating the N-fused pyrroles under visible light photocatalytic conditions. The use of K-PHI as heterogeneous photocatalyst guarantees the recycling and sustainability of the whole transformation, and there is no obvious impact on the conversion and selectivity after recycling K-PHI for three times. A variety of different substitution patterns are accessible through the manifold presented by the authors. Furthermore, in most cases the yields range from good to excellent with excellent regioselectivity.

Conceptually, this work is one of few examples of photocatalytic tandem organic reactions that enables regioselective coupling of the protonated aminium radicals with the carbon atom of the C=O group in α,β -unsaturated ketones. The procedure involves the first photocatalytic C-N bond formation and the second C-C bond formation accompanied by the elimination of H₂ and H₂O molecules. The authors have carefully investigated the catalytic process by cyclic voltammetry study, EPR spectroscopy, spin population analysis and DFT calculations. Overall, this is very nice work, and merits publication in Nat. Commun..

I would ask the authors to consider the following very small considerations:

1. Only diaryl-substituted chalcones were utilized under the photoredox reaction conditions, the use of aliphatic enones would benefit the chemistry. What would take place if replace one or both of the aryl groups with alkyl group?

2. The compounds in Figure 1 are barely isoquinolines and can not describe the significance of 5,6-dihydropyrrolo[2,1-a]isoquinolines, which are fused heterocycles.

3. If it can be used to late-functionalization or the synthesis of one or two value-added chemicals, it would be more interesting.

Mechanism to discuss:

As to the mechanism, it is great to isolate isoquinolin-3-ol intermediate and isoquinolin-10b-d-3-ol calculate the process with DFT. There are other possibilities:

1. Is it possible to go through dehydration of amine and ketone, and then the C-C bond formation via intramolecular cyclization.

2. Is it possible to goes through Aza-Henry reaction to construct C-C bond, then dehydration reaction.

Reviewer #2 (Remarks to the Author):

Recommendation: Publish elsewhere.

Comments:

The manuscript by Bogdan Kurpil et al. describes a tandem photocatalytic reaction between tetrahydroisoquinoline (THIQ) and chalcones catalyzed by KPHI as a reusable photocatalyst. The reaction setup is simple and several N-fused pyrrole products have been synthesized in reasonable yields. DFT calculations as well as a few experiments have been conducted to investigate the reaction mechanism. To my knowledge and based on a literature search, this photocatalytic tandem reaction is previously unreported and interesting to organic chemists. The reaction involves two steps: 1) C-N formation between tetrahydroisoquinoline and chalcones to produce the aminoalcohol intermediate 6; in this part, the concept of a photocatalytic selective addition of N-centered radical to carbonyls is raised but lack of solid evidence; this step could be simply achieved by nucleophilic 1,2-addition as shown in Scheme 1 provided here; 2) C-C bond formation followed by elimination of H₂ and H₂O molecules, which is an extension of classical alpha-C-H oxidations of tertiaryamines. The article may be suitable for an organic journal or Communication Chemistry but not for consideration in a high ranked journal like Nature Communication. Acceptance of this manuscript is not recommended as it is not novel and groundbreaking enough. Here are the major reasons:

- 1) This study is still an extension of photocatalytic transformations of N-substituted THIQ. Even though a concept "photocatalytic selective addition of N-centered radical to carbonyls" has been raised, the universality of the proposed concept has not well-demonstrated, especially in term of the mechanism, broad reaction scopes and their practical useful.
- 2) The reaction-scopes have been very poorly studied. For example, only one amine-type substrate, tetrahydroisoquinoline, was tested in this photocatalytic tandem reaction. Can the reaction-scope be extended to other tetrahydroisoquinoline derivatives and other alpha, beta-unsaturated ketones? Otherwise, the reaction-scopes will be really-limited.
- 3) There is still lack of solid evidences to support their key concept/mechanism, the selective addition of N-centered radical to carbonyls. Several radical intermediate species have been proposed. However, no experimental results such as in-situ ESR or in-situ NMR are performed to demonstrate the presence of these radical species and the formation of the proposed intermediates.
- 4) The authors claim that the selective addition of N-centered radical to carbonyls is the key step for the C-N formation between tetrahydroisoquinoline and chalcones to produce the hemi-aminal intermediate 6. However, the C-N formation between tetrahydroisoquinoline and chalcones may also undergo very classical and simple nucleophilic 1,2-addition reaction as widely proposed in previous studies (i. e. *Org. Lett.*, 2012, 5656-5659). In the literature, the nucleophilic 1,2-addition of secondary amine to alpha, beta-unsaturated ketones have been well demonstrated and the hemi-aminal intermediate can be readily formed; the radical addition of secondary amine to carbonyl is clearly not involved. Since the manuscript raised the new concept of regioselective aminium radical addition to the carbonyl bond for the C-N formation, solid evidences to capture the active radical species are very necessary; The classical reaction pathway/mechanism, nucleophilic 1,2-addition of secondary amine to alpha, beta-unsaturated ketones should be carefully examined (scheme 1).

[Redacted]

5) The reactions are carried out in very small-scale (50 mmol) with highly weight-loading of photocatalysts (around 50 wt% catalyst-loading in term of the weight amount of chalcones). Large-scale reactions (i. e. 1g-scale) with low catalyst loading should be addressed.

Other minor revisions are also required:

1) All of biological active derivatives of THIQ in figure 1 are N-substituted THIQ. The skeletons of these biological active derivatives in figure could not be synthesized by the strategy proposed in this manuscript. Thus, the Figure 1 contributes little to the true merit of the work or its technical content.

2) The NOESY spectrum of DHPIQ 3c is the evidence of structural correctness rather than the regioselectivity bond-formation between the THIQ N-atom and carbonyl group of the chalcone. The spectrum is more suitable for supporting information.

3) Control experiments without light or without photocatalyst should be re-studied in the presence of acetone.

4) In page 7, "only DHPIQ 3a was previously described in the literature, presumably due to the inapplicability of the existing methods for preparation of other derivatives". The statement is improper and groundless.

5) The format of the reference document is in a mess, such as missing page number in refs.45, 49; unnecessary number in ref.9 and letters in ref.51.

Reviewer #3 (Remarks to the Author):

The authors demonstrated high yields of 1,3-disubstituted-5,6-dihydropyrrolo[2,1-a]isoquinolines (DHPIQ) from the photocatalytic reaction between tetrahydroisoquinoline (THIQ) and chalcones to form under visible light irradiation without the use of additives. This is achieved used potassium poly(heptazine imides) (K-PHI) as a photocatalyst. While many papers have performed photocatalytic transformations of THIQ, they tend to be limited to the C2-atom functionalization of N-substituted THIQ. This reaction is interesting as it enables the one-pot functionalization of two reactive centers in THIQ, the C2 atom and the N atom, by the C=C and C=O functional groups of chalcones. The authors have made a comprehensive study of the reaction mechanism. This manuscript is recommended for acceptance after some questions are addressed.

1. The authors claimed that K-PHI was the photocatalyst of choice as its negatively charged nitrogen atoms resulted in a band structure with a more positive valence band that is suitable for oxidation reactions. It is suggested that control experiments should be done with a series of other carbon nitride photocatalysts to demonstrate the superiority of K-PHI in terms of photocatalytic performance. It was also shown that Ir(ppy)₃ gave 89% selectivity. Is there an advantage that K-PHI possesses over Ir(ppy)₃, i.e., perhaps the surface properties of heterogeneous photocatalysts may grant it some

advantages over homogeneous photocatalysts (J. Am. Chem. Soc. 2017, 139, 269-276; Angew. Chem. Int. Ed. 2018, 57 (31), 9780-9784)?

2. It is suggested that the authors provide some discussion on the applications and significance of DHPIQ in the introduction. It is also suggested that the authors shorten the discussion on THIQ, i.e., Figure 1 may not be necessary.

3. It is suggested that the authors expand on the introduction on photocatalysis, i.e., the current breakthroughs and challenges of the field.

4. It would be better if the authors provided some sort of explanation on why using oxygen as an electron scavenger instead of acetone gave DHIQ instead of DHPIQ.

5. Can the R group of chalcones be replaced with other functional groups, such as aliphatic groups. If not, can an explanation be provided?

6. The authors mentioned that the delocalization of the uncoupled electron in the chalcone radical anion over the carbon atoms of C=O and the benzyl group may be due to the absence of Lewis acids that are otherwise coordinated to the oxygen bond of the chalcone. If the authors added Lewis acid to destabilize the chalcone radical, would the reaction not occur and would the EPR signal be different?

7. The authors mentioned that step h occurs via the thermal elimination of a H₂O molecule. In that case, would a lower temperature (i.e., <80 °C) cause the rate of product formation from 10 to be slower?

8. Can the chalcone be replaced by two molecules, one comprising C=C and the other comprising C=O?

9. The presentation of the manuscript may need to be improved. It is suggested that the authors merge short paragraphs of similar themes so that the reader can quickly locate the information they need. Can the data from the theoretical simulations also be made into figures for easy reference by readers?

Replies to the referees' comments to the manuscript "Photocatalytic Carbon Nitride Assisted Domino Reaction Yields N-fused Pyrroles via a Regioselective Aminium Radical Addition to the Carbonyl Bond"

by Bogdan Kurpil, Katharina Otte, Artem Mishchenko, Paolo Lamagni, Wojciech Lipinski, Nina Lock, Markus Antonietti and Aleksandr Savateev submitted to Nature Communication.

Reviewers' comments:

Reviewer #1 (Remarks to the Author):

The manuscript from Antonietti, Savateev and coworkers reported the potassium poly(heptazine imide) photocatalyzed aminium radical addition to the carbonyl bond for the synthesis of 1,3-disubstituted-5,6-dihydropyrrolo[2,1-a]isoquinolines (DHPIQ). Most N-centered radicals are derived from α -Amido-oxy, azirine, sulfonamide, hydrazine or hydrazone. There are very few reports about N-unsubstituted tetrahydroisoquinolines as a source of N-centered radicals. In this manuscript the authors show that NH-THIQ are viable coupling precursors with chalcones, generating the N-fused pyrroles under visible light photocatalytic conditions. The use of K-PHI as heterogeneous photocatalyst guarantees the recycling and sustainability of the whole transformation, and there is no obvious impact on the conversion and selectivity after recycling K-PHI for three times. A variety of different substitution patterns are accessible through the manifold presented by the authors. Furthermore, in most cases the yields range from good to excellent with excellent regioselectivity.

Conceptually, this work is one of few examples of photocatalytic tandem organic reactions that enables regioselective coupling of the protonated aminium radicals with the carbon atom of the C=O group in α,β -unsaturated ketones. The procedure involves the first photocatalytic C-N bond formation and the second C-C bond formation accompanied by the elimination of H₂ and H₂O molecules. The authors have carefully investigated the catalytic process by cyclic voltammetry study, EPR spectroscopy, spin population analysis and DFT calculations. Overall, this is very nice work, and merits publication in Nat. Commun..

Response: We are grateful the reviewer for a very positive evaluation of our work and recommendations for improvement the present manuscript. Below we provide point-by-point replies to the comments.

I would ask the authors to consider the following very small considerations:

1. Only diaryl-substituted chalcones were utilized under the photoredox reaction conditions, the use of aliphatic enones would benefit the chemistry. What would take place if replace one or both of the aryl groups with alkyl group?

Response: We have studied additionally a series of chalcones with aliphatic substituents.

Thus, chalcones **2l** and **2m** gave the respective DHPIQs, **3l** and **3m**, while (E)-hex-4-en-3-one (A) and (E)-4,4-dimethyl-1-phenylpent-2-en-1-one (B) did not react with THIQ. Such behavior might be explained by lower stability of the intermediary radical anions. This hypothesis was supported by cyclic voltammetry study on chalcones. Thus, methyl-substituted chalcones **2l,m** exhibit irreversible reduction peak at -1.45 V and -1.5 V vs. SCE, which are more negative compared to diaryl-substituted chalcones (Figure 3). *Tert*-butyl-substituted chalcone (B) showed reversible reduction peak at -1.67 V, implying that the subsequent reaction, e.g. coupling of the radicals, is inhibited and at the given scan rate of 50 mV·s⁻¹ radical anion is oxidized back to the chalcone. However, the bulkiness of *tert*-butyl substituent obstructs efficient condensation with THIQ. CV curve of A is flat in the range from -1.75 V to +1.5 V, suggesting that no redox process takes place in

this region, and reduction power of the photocatalyst is not sufficient to reduce aliphatic chalcone to the respective radical anion.

In general, substitution of aryl groups in chalcones with aliphatic substituents leads to destabilization of the intermediary radical anions and therefore shifts the reduction potential to more negative values, due to worse delocalization of uncoupled electron in these systems compared to those bearing only (het)aryl substituents in position 1 and 3.

In addition, as was suggested by the reviewer #3, the importance of adsorption of chalcones on the K-PHI surface, as a key step on the path to chalcone reduction, cannot be excluded. Thus, charge transfer from K-PHI⁻ to flat and rigid diarylsubstituted chalcones, due to π - π interaction between heptazine unit and aromatic group, is more efficient compared to one- or di-alkylsubstituted chalcones.

The results and discussion were added to the manuscript.

2. The compounds in Figure 1 are barely isoquinolines and can not describe the significance of 5,6-dihydropyrrolo[2,1-a]isoquinolines, which are fused heterocycles.

Response: The similar comment was provided by all reviewers. Therefore, Figure 1 and related discussion was removed from the manuscript.

3. If it can be used to late-functionalization or the synthesis of one or two value-added chemicals, it would be more interesting.

Response: We have revised the literature available on 5,6-dihydropyrrolo[2,1-a]isoquinolines (DHPIQs). Currently there are mainly two areas where these heterocycles are known or applied. Firstly, DHPIQ fragment is incorporated into the structure of natural compounds called lamellarins.[*Angew Chem* 2004, **116**(7): 884-886] Secondly, polyarylsubstituted DHPIQs were suggested for use "in electronic devices including organic photovoltaic cells, organic field effect transistors (OFETs), and light emitting devices".[patent WO2018058493A1]

We have performed basic spectroscopic characterization of some DHPIQs synthesized in this work in order to show possible application of these molecules as value added compounds, for example, in organic photovoltaics. These compounds exhibit pronounced blue fluorescence. The results are summarized in Figure 4. Relevant discussion was added to the manuscript.

Mechanism to discuss:

As to the mechanism, it is great to isolate isoquinolin-3-ol intermediate and isoquinolin-10b-d-3-ol calculate the process with DFT.

There are other possibilities:

1. Is it possible to go through dehydration of amine and ketone, and then the C-C bond formation via intramolecular cyclization.

Response: In order to check this possibility, we have performed a reaction between dihydroisoquinoline and 1,3-diphenylprop-2-yn-1-one under the conditions similar to DHPIQ synthesis (acetonitrile, 80°C, N₂, 20 h, 461 nm). In this case no DHIQ was formed. Results were added to the ESI.

2. Is it possible to go through Aza-Henry reaction to construct C-C bond, then dehydration reaction.

Response: We have performed an experiment between dihydroisoquinoline and chalcone. In this case, also no DHPIQ was formed. Results were added to the ESI.

Reviewer #2 (Remarks to the Author):

Recommendation: Publish elsewhere.

Comments:

The manuscript by Bogdan Kurpil et al. describes a tandem photocatalytic reaction between tetrahydroisoquinoline (THIQ) and chalcones catalyzed by KPHI as a reusable photocatalyst. The reaction setup is simple and several N-fused pyrrole products have been synthesized in reasonable yields. DFT calculations as well as a few experiments have been conducted to investigate the reaction mechanism. To my knowledge and based on a literature search, this photocatalytic tandem reaction is previously unreported and interesting to organic chemists. The reaction involves two steps: 1) C-N formation between tetrahydroisoquinoline and chalcones to produce the aminoalcohol intermediate 6; in this part, the concept of a photocatalytic selective addition of N-centered radical to carbonyls is raised but lack of solid evidence; this step could be simply achieved by nucleophilic 1,2-addition as shown in Scheme 1 provided here; 2) C-C bond formation followed by elimination of H₂ and H₂O molecules, which is an extension of classical alpha-C-H oxidations of tertiary amines. The article may be suitable for an organic journal or Communication Chemistry but not for consideration in a high ranked journal like Nature Communication. Acceptance of this manuscript is not recommended as it is not novel and groundbreaking enough.

Response: We appreciate the time that referee spent to evaluate our work. After revision we have expanded the scope of chalcones that can be used in synthesis of DHPIQ to those bearing not only aromatic and heteroaromatic substituents, we have shown that aliphatic enones may be also used to obtain the corresponding DHPIQs. We have also expanded the scope of THIQs that can be used as components in this photocatalytic reaction. The possibility to perform synthesis using much larger loadings of substrates was proved. Additional, in situ EPR and NMR were performed. The results support the proposed mechanism. Basic optical properties of DHPIQs were studied, suggesting that these heterocycles might be useful in organic optoelectronics. Below we provide point by point replies to the referee's comments.

Here are the major reasons:

1) This study is a still an extension of photocatalytic transformations of N-substituted THIQ. Even though a concept "photocatalytic selective addition of N-centered radical to carbonyls" has been raised, the universality of the proposed concept has not well-demonstrated, especially in term of the mechanism, broad reaction scopes and their practical useful.

Response: Regarding the reaction scope. After the revision, the manuscript reports 15 DHPIQs plus 2 reduced DHPIQs **7p** and **7p-d₁**. We have shown that this photocatalytic reaction works for different readily available chalcones:

- 1) 1,3-diarylsubstituted chalcones (compounds **3a-h**)
- 2) chalcones bearing one aromatic and one heteroaromatic substituent (compounds **3i-k**)
- 3) chalcones bearing at least one alkyl substituent, either in 1st or 3rd position (compounds **3l** and **3o**)
- 4) 6-fluoro and 6-trifluoromethyl-substituted tetrahydroisoquinolines also gave the corresponding DHPIQs.

We have performed additional experiments in order to confirm the mechanism. Thus, in situ NMR suggested that in the absence of light and photocatalyst, no DHPIQ was formed, while THIQ was added to the double bond of chalcone. This process is known as aza-Michael reaction (also see reply to the comment 4). [Advanced Synthesis & Catalysis, 2005, 347(6), 763-766], [Synthetic Communications, 2010, 40, 1730-1735], [Inorganica Chimica Acta, 2005, 358(9), 2749-2754] etc.

In situ EPR confirmed reductive quenching of the excited state of K-PHI* - a signal of K-PHI^{•-} was detected.

Taking into account oxidation potentials of the components present in the reaction mixture, THIQ ($E_{\text{ox}} = +0.8$ V vs. SCE), chalcone ($E_{\text{ox}} > +1.5$ V vs. SCE), acetonitrile ($E_{\text{ox}} > +1.5$ V vs. SCE), potentials of the valence band ($E_{\text{VB}} = +1.96$ V vs. SCE) and conduction band ($E_{\text{CB}} = +1.96$ V vs. SCE) of K-PHI, it can be concluded that THIQ works as electron donor, because it has the lowest oxidation potential among the present in the reaction mixture chemical compounds. Single electron oxidation of THIQ in turn leads to the radical cation. This step in the photocatalytic mechanism was also suggested on the basis of a number of articles, where amines were employed as electron donors in photocatalysis. [J Am Chem Soc 2016, **138**(36): 11860-11871. (Scheme 3, therein)], [Chem Commun 2011(47): 12709-12711 (Scheme 1, therein)], [Chem Eur J 2012, **18**(17): 5170-5174 (Scheme 2, therein)], [Green Chem 2011(13): 3341-3344 (Figure 2, therein)], [J Fluorine Chem 2012, **140**: 88-94 (Scheme 2, therein)], [Chem. Commun., 2014, 50, 15593 - Figure 3 therein].

Using the optimized conditions (Table 5), the reaction can be scaled, for example, to 300 mg of the chalcone **2a**. In this case the yield of DHPIQ **3a** was 67%. We attribute this to higher optical density of the reaction mixture and hence worse delivery of light to the photocatalyst particles in the bulk of the reactor.

2) The reaction-scopes have been very poorly studied. For example, only one amine-type substrate, tetrahydroisoquinoline, was tested in this photocatalytic tandem reaction. Can the reaction-scope be extended to other tetrahydroisoquinoline derivatives and other alpha, beta-unsaturated ketones? Otherwise, the reaction-scopes will be really-limited.

Response: After the revision, the manuscript reports 15 DHPIQs. We have shown that this photocatalytic reaction works for different substrates:

- 1) 1,3-diarylsubstituted chalcones (compounds **3a-h**)
- 2) chalcones bearing one aromatic and one heteroaromatic substituent (compounds **3i-k**)
- 3) chalcones bearing at least one alkyl substituent, either in 1st or 3rd position (compounds **3l** and **3o**)
- 4) in terms of variation of tetrahydroisoquinoline core, 6-fluoro and 6-trifluoromethyl-substituted compounds also gave the corresponding DHPIQs.

These results and discussion were added to the manuscript.

3) There is still lack of solid evidences to support their key concept/mechanism, the selective addition of N-centered radical to carbonyls. Several radical intermediate species have been proposed. However, no experimental results such as in-situ ESR or in-situ NMR are performed to demonstrate the presence of these radical species and the formation of the proposed intermediates.

Response: According to the referee's suggestion we have performed in-situ EPR. The tube for EPR analysis was charged with a suspension of K-PHI and a mixture of reagents, placed into the EPR spectrometer and EPR spectrum was acquired while sample was either in dark or irradiated with blue (461 nm, 6 mW·cm⁻²) light. In EPR spectra, shown below (please note that EPR spectrum recorded in dark was magnified 20 times for clarity of representation), we detected a signal of the reduced photocatalyst – K-PHI⁻. The intensity of the signal was drastically enhanced when in situ EPR was performed under light irradiation. This experiment suggests that reductive quenching of K-PHI* excited state by amine ($E_{ox} = +0.8$ V vs. SCE) takes place.

The concept of ammonium radical addition to the carbonyl group of chalcone is proposed based on data (also see replies to comments 1 and 4):

1. Without photocatalyst and in dark amine is added to the C=C bond of the α,β -unsaturated ketone (aza-Michael reaction, NMR spectra of cyclohexanone + THIQ and chalcone + THIQ below).

2. Reductive quenching of the photocatalyst is implemented (EPR spectra). Among all components present in the reaction mixture, THIQ has the lowest oxidation potential. Therefore it is the most probable electron donor. In order to comply with balance of electrons, THIQ is oxidized to the radical cation.

3. DFT calculations suggest that uncoupled electron is located at the N-atom of THIQ, while positive charge is localized on hydrogen atom directly bound to N-atom.

4. Oxidation of amines to the radical cations, was reported by a number of research groups earlier; however, the intermediary radical cations in these works were not detected either (including EPR). [*J Am Chem Soc* 2016, **138**(36): 11860–11871. (Scheme 3, therein)], [*Chem Commun* 2011(47): 12709-12711 (Scheme 1, therein)], [*Chem Eur J* 2012, **18**(17): 5170–5174 (Scheme 2, therein)], [*Green Chem* 2011(13): 3341-3344 (Figure 2, therein)], [*J Fluorine Chem* 2012, **140**: 88-94 (Scheme 2, therein)], [*Chem. Commun.*, 2014, 50, 15593 – Figure 3 therein]. The absence of other species might possibly be explained by short lifetime and their low concentration.

4) The authors claim that the selective addition of N-centered radical to carbonyls is the key step for the C-N formation between tetrahydroisoquinoline and chalcones to produce the hemi-aminal intermediate 6. However, the C-N formation between tetrahydroisoquinoline and chalcones may also undergo very

classical and simple nucleophilic 1,2-addition reaction as widely proposed in previous studies (i. e. *Org. Lett.*, 2012, 5656-5659). In the literature, the nucleophilic 1,2-addition of secondary amine to α,β -unsaturated ketones have been well demonstrated and the hemi-aminal intermediate can be readily formed; the radical addition of secondary amine to carbonyl is clearly not involved. Since the manuscript raised the new concept of regioselective aminium radical addition to the carbonyl bond for the C-N formation, solid evidences to capture the active radical species are very necessary; The classical reaction pathway/mechanism, nucleophilic 1,2-addition of secondary amine to α,β -unsaturated ketones should be carefully examined (scheme 1).

Response: Indeed, nucleophilic addition of amine to the carbonyl group might appear as a very obvious and intuitive process. We are grateful the referee for raising the question about addition of secondary amines to α,β -unsaturated compounds, because it requires explanation. The article *Org. Lett.*, 2012, 5656-5659, mentioned by the referee, does not report nucleophilic addition of piperidine to cyclohex-2-en-1-one, neither manuscript itself nor supporting information. Please provide page number and Scheme/Figure number, if the reaction is, in fact, reported in this article.

In addition, we have not found such reaction reported in any other document. As on 11.10.2018, no results were found in SciFinder database:

[Redacted]

Finally, in order to check if cyclohexanone and piperidine in principle can react as was suggested by the reviewer, we did in situ NMR experiment (cyclohexanone 0.2 mmol, piperidine 0.2 mmol, C_6D_6 0.4 mL, room temperature, 12h). As can be seen from the 1H and ^{13}C NMR spectra below, piperidine was not added to the carbonyl group. Instead, aza-Michael addition occurred:

In addition, there are reports in the literature, when α,β -unsaturated carbonyl compounds are allowed to react with secondary amines, aza-Michael reaction is successfully implemented.[Advanced Synthesis & Catalysis, 2005, 347(6), 763-766 – Table 2, entry 4 (cyclohexanone + piperidine) therein], [Synthetic Communications, 2010, 40, 1730-1735 – Table 1, entry 6 (cyclohexanone + piperidine) therein], [Inorganica Chimica Acta, 2005, 358(9), 2749-2754 – Table 2, entry 15 (cyclohexanone + piperidine) therein] etc.

In the present study, when THIQ and chalcone **2a** were mixed together in NMR tube (THIQ 150 μmol , chalcone **2a** 50 μmol , CD_3CN 0.4 mL, room temperature, 24 h) amine was not added to the chalcone carbonyl group. Instead we observed aza-Michael addition of THIQ to the $\text{C}=\text{C}$ double bond of the chalcone ($\sim 57\%$). [Org.Lett. 2014, 16, 3158–3161 – please see Equation 3 in this article and Page S70 of the ESI for NMR spectra]

Chalcone **2d** reacted with THIQ in dark in a similar fashion. The Michael adduct ($\sim 31\%$) was detected by ^1H NMR.

An attempt to convert aza-Michael adduct into the corresponding DHPIQ, using the standard conditions, was not successful (see ^1H NMR of the reaction mixture below).

This experiment proves that aza-Michael adduct is not the intermediate of DHPIQ formation. Summarizing the all abovementioned NMR experiments, addition of THIQ to the carbonyl group does not proceed in dark and without K-PHI. Under these conditions, however, Michael addition of THIQ to the double bond of chalcone is implemented. For clarity these experiments are summarized on the scheme below and also in the ESI.

In order to probe the reaction mixture for the presence of radical species we have performed in situ EPR. In this experiment the EPR tube was loaded with K-PHI and a mixture of reagents in MeCN, placed into the EPR spectrometer and EPR spectrum was acquired while sample was either in dark or irradiated with blue (461 nm, 6 mW·cm⁻²) light. In EPR spectra, shown below (please note that EPR spectrum recorded in dark was magnified 20 times in order to discern a signal), we detected K-PHI^{•-} radical anion signal. The intensity of the signal was drastically enhanced when in situ EPR was performed under light irradiation. It suggests that indeed the excited state of the photocatalyst is reductively quenched by amine. In order to obey the balance of electrons in this process, amine has to be converted into the radical cation.

The concept of ammonium radical addition to the carbonyl group of chalcone is proposed based on the data provided above:

1. Without photocatalyst and in dark amine is added only to the C=C bond of the α,β -unsaturated ketone (aza-Michael reaction, NMR spectra of cyclohexanone + THIQ and chalcone + THIQ above).
2. Reductive quenching of the photocatalyst is implemented (EPR spectra). Among all components present in the reaction mixture, THIQ has the lowest oxidation potential (+0.8 V vs SCE determined by CV). Therefore it is the most probable electron donor.
3. DFT calculations suggest that uncoupled electron is located at the N-atom of THIQ, while positive charge is localized on hydrogen atom directly bound to N-atom.
4. Oxidation of amines to the radical cations, was reported by a number of research groups earlier; however, the intermediary radical cations in these works were not detected either (including EPR). [*J Am Chem Soc* 2016, **138**(36): 11860–11871. (Scheme 3, therein)], [*Chem Commun* 2011(47): 12709-12711 (Scheme 1, therein)], [*Chem Eur J* 2012, **18**(17): 5170–5174 (Scheme 2, therein)], [*Green Chem* 2011(13): 3341-3344 (Figure 2, therein)], [*J Fluorine Chem* 2012, **140**: 88-94 (Scheme 2, therein)], [*Chem. Commun.*, 2014, 50, 15593 – Figure 3 therein]. The absence of other species might possibly be explained by short lifetime and their low concentration.

The scheme of the tentative mechanism and discussion was modified taking into account the results obtained during the manuscript revision.

5) The reactions are carried out in very small-scale (50 mmol) with highly weight-loading of photocatalysts (around 50 wt% catalyst-loading in term of the weight amount of chalcones). Large-scale reactions (i. e. 1g-scale) with low catalyst loading should be addressed.

Response: In general, in heterogeneous catalysis the reaction occurs on the surface of the catalyst. Typically, material in bulk is not involved in catalysis. This is very different from homogeneous catalysis, in which every molecule of the catalyst present in the solution, in principle, can enable chemical transformation.

The heterogeneous photocatalyst used in the present study, K-PHI, is represented by particles with average particle size determined by DLS 100 nm (Figure S1).

From N_2 sorption isotherm measured at 77K we deduced that (1) K-PHI is non-porous material (Type II isotherm) and (2) it has specific surface area $89 \text{ m}^2 \text{ g}^{-1}$ (Figure S1). Therefore only external surface of K-PHI particles is exposed to the solution of reagents and contributes to the photocatalytic activity.

From powder X-Ray diffractogram and TEM image (Figure S1) it is seen that K-PHI is rather well ordered crystalline material. It has layered structure with a distance between layers 0.3 nm, while each layer consists of repeating heptazine units.[Adv. Mater., 2017, 29(32), 1700555, doi: 10.1002/adma.201700555] Knowing the structure of K-PHI we can estimate the volume occupied by one heptazine unit (the smallest photocatalytic site) – 0.141 nm^3 . The number of heptazine units in the cubic particle (with the length of the cube edge 100 nm that was determined by DLS) is $7 \cdot 10^7$, while the number of heptazine units on the surface of the same cubic particle is $9 \cdot 10^3$. In other words, only one out of 800 heptazine units is located on the surface of the particle. Using these calculations we can deduce that even though 50 wt. % of K-PHI were taken for catalysis, the real amount of “active” material does not exceed 1 wt. %.

The price of carbon nitride was calculated to be 0.1 Euro/g.[Y. Dai et al. *Nature Communications* 2018, 9(1): 60.] Given that K-PHI can be easily recycled (Table 1, entries 14-16) and used again, the corrected price is even lower.

Using the optimized conditions (Table 5), the reaction can be scaled, for example, to 300 mg of the chalcone **2a**. In this case the yield of DHPIQ **3a** was 67%. We attribute this to higher optical density of the reaction mixture and hence worse delivery of light to the photocatalyst particles in the bulk of the reactor.

Other minor revisions are also required:

1) All of biological active derivatives of THIQ in figure 1 are N-substituted THIQ. The skeletons of these biological active derivatives in figure could not be synthesized by the strategy proposed in this manuscript. Thus, the Figure 1 contributes little to the true merit of the work or its technical content.

Response: The similar comment was provided by all reviewers. Therefore, Figure 1 and related discussion was removed from the manuscript.

2) The NOESY spectrum of DHPIQ 3c is the evidence of structural correctness rather than the regioselectivity bond-formation between the THIQ N-atom and carbonyl group of the chalcone. The spectrum is more suitable for supporting information.

Response: The sentence was rephrased.

3) Control experiments without light or without photocatalyst should be re-studied in the presence of acetone.

Response: As suggested by the referee, the control experiments without photocatalyst and without light irradiation, both in the presence of acetone were performed. In both cases, no DHPIQ was formed. The results were added to the Table 1 (entries 6 and 7).

4) In page 7, “only DHPIQ 3a was previously described in the literature, presumably due to the inapplicability of the existing methods for preparation of other derivatives”. The statement is improper and groundless.

Response: The statement was removed from the manuscript.

5) The format of the reference document is in a mess, such as missing page number in refs.45, 49; unnecessary number in ref.9 and letters in ref.51.

Response: All references were checked and corrected.

Reviewer #3 (Remarks to the Author):

The authors demonstrated high yields of 1,3-disubstituted-5,6-dihydropyrrolo[2,1-a]isoquinolines (DHPIQ) from the photocatalytic reaction between tetrahydroisoquinoline (THIQ) and chalcones to form under visible light irradiation without the use of additives. This is achieved used potassium poly(heptazine imides) (K-PHI) as a photocatalyst. While many papers have performed photocatalytic transformations of THIQ, they tend to be limited to the C2-atom functionalization of N-substituted THIQ. This reaction is interesting as it enables the one-pot functionalization of two reactive centers in THIQ, the C2 atom and the N atom, by the C=C and C=O functional groups of chalcones. The authors have made a comprehensive study of the reaction mechanism. This manuscript is recommended for acceptance after some questions are addressed.

Response: We appreciate very much positive evaluation of our manuscript by the referee, suggestions and attention to the details aiming to improve further the quality of this work. Below we provide point-by-point replies to the comments.

1. The authors claimed that K-PHI was the photocatalyst of choice as its negatively charged nitrogen atoms resulted in a band structure with a more positive valence band that is suitable for oxidation reactions. It is suggested that control experiments should be done with a series of other carbon nitride photocatalysts to demonstrate the superiority of K-PHI in terms of photocatalytic performance. It was also shown that Ir(ppy)₃ gave 89% selectivity. Is there an advantage that K-PHI possesses over Ir(ppy)₃, i.e., perhaps the surface properties of heterogeneous photocatalysts may grant it some advantages over homogeneous photocatalysts (J. Am. Chem. Soc. 2017, 139, 269-276; Angew. Chem. Int. Ed. 2018, 57 (31), 9780-9784)?

Response: We have tested mesoporous graphitic carbon nitride (mpg-CN), a carbon nitride material with surface area 200 m²/g, and sodium poly(heptazine imide) (Na-PHI), a related to K-PHI material, in photocatalytic synthesis of DHPIQ **3a**. Under the similar conditions, developed for K-PHI, mpg-CN gave 35% conversion of chalcone **2a**, while DHPIQ **3a** was obtained with 81% selectivity. Na-PHI converted 85% of chalcone **2a** and DHPIQ **3a** was obtained with 93% selectivity. The relevant discussion was added to the manuscript.

We agree that the surface properties of the heterogeneous photocatalyst do contribute to the observed activity. The importance of the surface properties in heterogeneous photocatalysis was highlighted in the introduction and aforementioned articles were referred.

2. It is suggested that the authors provide some discussion on the applications and significance of DHPIQ in the introduction. It is also suggested that the authors shorten the discussion on THIQ, i.e., Figure 1 may not be necessary.

Response: The similar comment was provided by all reviewers. Therefore, Figure 1 and related discussion was removed from the manuscript.

In addition, we revised the literature available on 5,6-dihydropyrrolo[2,1-a]isoquinolines (DHPIQs). Currently there are mainly two areas where these heterocycles are known or applied. Firstly, DHPIQ fragment is incorporated into the structure of natural compounds called lamellarins.[*Angew Chem* 2004, **116**(7): 884-886] Secondly, polyarylsubstituted DHPIQs were suggested for use “in electronic devices including organic photovoltaic cells, organic field effect transistors (OFETs), and light emitting devices”.[patent WO2018058493A1]

We have performed basic spectroscopic characterization of some DHPIQs synthesized in this work. DHPIQs exhibit pronounced blue fluorescence, which might be useful, for example, in organic photovoltaics. The results are summarized in Figure 4. Relevant discussion was added to the manuscript.

3. It is suggested that the authors expand on the introduction on photocatalysis, i.e., the current breakthroughs and challenges of the field.

Responses: The introduction section was modified in order to show the readers progress in the field.

4. It would be better if the authors provided some sort of explanation on why using oxygen as an electron scavenger instead of acetone gave DHIQ instead of DHPIQ.

Response: According to the proposed mechanism, on the first step chalcones work as electron acceptors. The estimated from CV curves reduction potential of chalcones, depending on the structure, ranges from -1.0 V up to -1.5 V vs. SCE.

Standard electrode potential of the reaction $O_2 + 1e = O_2^{\cdot -}$ is -0.57 V vs. SCE, $O_2 + H^+ + 1e = HO_2^{\cdot}$ is -0.37 V vs. SCE, $O_2 + H_2 + 2e = H_2O_2$ is +0.46 V vs. SCE. Therefore reduction of oxygen to any of these species is more favorable process compared to the reduction of chalcone. In other words, when both oxygen and chalcone are present in the reaction mixture, oxygen is more competitive to take the electron from K-PHI radical anion.

THIQ can be oxidized photocatalytically in the presence of O_2 to dihydroisoquinoline (DHIQ). [Angew. Chem. Int. Ed. **2010**, 50 (3), 657-660; ACS Catalysis **2016**, 6, 2754-2759] We have shown that DHIQ is not the intermediate in DHPIQ synthesis, because DHPIQ was not obtained when DHIQ was mixed with chalcone or with 1,3-diphenylprop-2-yn-1-one. Please see the response to the comments of the 1st reviewer regarding mechanism investigation.

The role of acetone in this reaction is to accept a molecule of hydrogen. Without acetone, chalcone works as acceptor of hydrogen. This, in turn, decreases the yield of DHPIQ (Table 1, entry 3).

5. Can the R group of chalcones be replaced with other functional groups, such as aliphatic groups. If not, can an explanation be provided?

Response: We have expanded the scope of chalcones to those having methyl-group either at 1st or at 3rd position.

Both chalcones, compounds **2l** and **2m**, gave the corresponding DHPIQs **3l** and **3m** with 50% and 15% respectively. However, (E)-hex-4-en-3-one (A) and tert-butylsubstituted chalcone (B) did not react with THIQ. The possible explanation of the inertness of these aliphatic chalcones might be that intermediary radical anions are not stable, i.e. there is no possibility for stabilization via conjugation with aromatic/heteroaromatic ring. This hypothesis was confirmed by CV. Because of the presence of alkyl group, chalcone **2l** has reduction potential shifted to ca. -1.45 V vs. SCE, while chalcone **2m** has a reduction potential -1.5 V vs. SCE. These results underline the importance of the aromatic group in position 3 of the chalcone. CV curve of chalcone A is flat in the range of scanned potentials (from -1.75 V to +1.5 V vs. SCE). Therefore reduction power of the photocatalyst is not sufficient to reduce this chalcone. Unlike to the rest of chalcones, tert-butyl-substituted chalcone showed reversible redox peak at -1.67 V, which implies that the subsequent reaction, e.g. coupling of the radicals, is inhibited and at the given scan rate of $50 \text{ mV} \cdot \text{s}^{-1}$ radical anion is oxidized back to the chalcone. All these findings were added to the manuscript.

6. The authors mentioned that the delocalization of the uncoupled electron in the chalcone radical anion over the carbon atoms of C=O and the benzyl group may be due to the absence of Lewis acids that are otherwise coordinated to the oxygen bond of the chalcone. If the authors added Lewis acid to destabilize the chalcone radical, would the reaction not occur and would the EPR signal be different?

Response: We have performed an experiment in the presence of AlCl_3 (1 eq vs chalcone) as a Lewis acid. In this case we observed 81% conversion of the chalcone **2a** and 91% of selectivity toward DHPIQ **3a**, which is slightly lower compared to the standard conditions (93% conversion, 89% selectivity, entry 5, Table 1). In this reaction in the presence of secondary amine, Lewis acid is apparently coordinated by THIQ rather than chalcone. Previous reports [*Science* 2016, **354**(6318): 1391-1395; *Angew Chem Int Ed* 2017, **56**(39): 11891-11895] focused on using chalcones in [2+2] cycloaddition, where Lewis acids additives were used, the reaction mixture did not contain amines, therefore coordination of carbonyl group to metal center can be realized.

7. The authors mentioned that step h occurs via the thermal elimination of a H_2O molecule. In that case, would a lower temperature (i.e., <80 oC) cause the rate of product formation from **10 to be slower?**

Response: Good conversion (85%) and selectivity ($\sim 85\%$) in this reaction can be achieved also at room temperature (entry 13, Table 1). However, the reaction time should be extended from 20 h to 60 h.

8. Can the chalcone be replaced by two molecules, one comprising C=C and the other comprising C=O?

Response: In order to check this possibility, we have performed following experiments:

1) Styrene (50 μmol) + benzaldehyde (50 μmol) [THIQ (3eq., 150 μmol), K-PHI (5mg), acetone (3eq., 9mg) CH_3CN (2 ml), $\lambda=461\text{nm}$, argon atmosphere] - Conversion: 0%; Selectivity: 0%.

2) Acetophenone (50 μmol) + benzaldehyde (50 μmol) [THIQ (3eq., 150 μmol), K-PHI (5mg), acetone (3eq., 9mg) CH_3CN (2 ml), $\lambda=461\text{nm}$, argon atmosphere] - Conversion: 0%; Selectivity: 0%.

In both cases no DHPIQ **3a** was obtained.

The results of these experiments suggest, that presence of chalcone – the molecule comprising two reactive centers, C=C and C=O bonds, is important in order to enable the reaction in a consecutive fashion.

9. The presentation of the manuscript may need to be improved. It is suggested that the authors merge short paragraphs of similar themes so that the reader can quickly locate the information they need. Can the data from the theoretical simulations also be made into figures for easy reference by readers?

Response: The manuscript was thoroughly revised. Theoretical calculations were separated from the mechanism discussion into a separate figure.

Reviewers' comments:

Reviewer #1 (Remarks to the Author):

The authors have adequately addressed the concerns raised and the quality of the manuscript has been improved after the revisions. It is publishable in the current shape.

Reviewer #2 (Remarks to the Author):

The manuscript by Bogdan Kurpil et al. describes a tandem photocatalytic reaction between tetrahydroisoquinoline (THIQ) and chalcones catalyzed by KPHI as a reusable photocatalyst. The highlight of this article is N-unsubstituted tetrahydroisoquinolines as a source of N-centered radicals. DFT calculations, cyclic voltammetry study, EPR spectroscopy and spin population analysis have been conducted to prove the key concept/mechanism. This manuscript has been very carefully revised and significantly improved. Therefore, it could be recommended for acceptance after major revision.

1. The C-N formation between tetrahydroisoquinoline and chalcones may also undergo very classical and simple nucleophilic 1,2-addition reaction as proposed in previous studies (i. e. Chaojun Li etc. *Org. Lett.*, 2012, 5606-5609). In the literature, the nucleophilic 1,2-addition of secondary amine to alpha, beta-unsaturated ketones have been well demonstrated and the hemi-aminal intermediate can be readily formed (*Org. Lett.*, 2012, 5606-5609. Page 5607 and SI-S7). The author did almost the same experiment but obtained very different results and conclusion. An explanation is definitely required.

[Redacted]

2. Page 4, 108-111: Although THIQ-1,2-d₂ have been used as substrate, a mixture of d-labeled ketones 4a-d₁ is obtained, which will make the reader very confuse to understand the reaction pathway. To this end, THIQ-1,1-d₂ is suggested as an alternative substrate.

3. The reaction products may be a class of useful materials, for example, in organic photovoltaics. The author can try to introduce active functional groups (such as bromine, alkyne etc) into chalcone or THIQ, allowing further functionalization of the polycyclic products.

4. The reaction requires three steps: (1) C-N bond formation between THIQ and the carbon atom of the C=O group; (2) C-C bond formation between THIQ and the chalcone; (3) elimination of a H₂ molecule and (4) elimination of a H₂O molecule. For the third step, the author is suggested to detect the generation of hydrogen by GC to identify whether it undergoes elimination of a H₂ molecule.

5. In the page 7, line 174,175, 9p should be changed to 7p. For deuterium-labeling experiment, the deuterium content should be clearly indicated.

Reviewer #3 (Remarks to the Author):

The authors conducted additional experiments to answer the questions in the comments. From my point of view, the questions about the reactions and mechanism have been well answered. The appropriate background and potential applications of this reaction have also been addressed in the manuscript. With these additional data and edited content, this article is acceptable in the current version.

Replies to the referees' comments to the manuscript "Photocatalytic Carbon Nitride Assisted Domino Reaction Yields *N*-fused Pyrroles *via* a Regioselective Aminium Radical Addition to the Carbonyl Bond" by Bogdan Kurpil, Katharina Otte, Artem Mishchenko, Paolo Lamagni, Wojciech Lipinski, Nina Lock, Markus Antonietti and Aleksandr Savateev submitted to Nature Communication.

Reviewer #1 (Remarks to the Author):

The authors have adequately addressed the concerns raised and the quality of the manuscript has been improved after the revisions. It is publishable in the current shape.

Response: We are grateful the referee for suggesting our manuscript for publication in its current shape.

Reviewer #2 (Remarks to the Author):

The manuscript by Bogdan Kurpil et al. describes a tandem photocatalytic reaction between tetrahydroisoquinoline (THIQ) and chalcones catalyzed by KPHI as a reusable photocatalyst. The highlight of this article is *N*-unsubstituted tetrahydroisoquinolines as a source of *N*-centered radicals. DFT calculations, cyclic voltammetry study, EPR spectroscopy and spin population analysis have been conducted to prove the key concept/mechanism. This manuscript has been very carefully revised and significantly improved. Therefore, it could be recommended for acceptance after major revision.

Response: We are grateful the referee for careful reading of the revised manuscript, reevaluation and the suggestions that would improve further the quality of our work. Point-by-point replies to the comments are given below.

1. The C-N formation between tetrahydroisoquinoline and chalcones may also undergo very classical and simple nucleophilic 1,2-addition reaction as proposed in previous studies (i. e. Chaojun Li et al. *Org. Lett.*, 2012, 5606-5609). In the literature, the nucleophilic 1,2-addition of secondary amine to α , β -unsaturated ketones have been well demonstrated and the hemiaminal intermediate can be readily formed (*Org. Lett.*, 2012, 5606-5609. Page 5607 and SI-S7). The author did almost the same experiment but obtained very different results and conclusion. An explanation is definitely required.

Response: The study by Chaojun Li reports synthesis of arylamines by oxidative aromatization of intermediary hemiaminals, which are the products of interaction between cyclic ketones, for example, cyclohexanone, and secondary amines.

There are several differences between the reaction reported by Chaojun Li et al and synthesis of DHPIQs (this work). In the photocatalytic approach of DHPIQs synthesis, chalcones are in *E*-configuration, while cyclohexenone, the model substrate studied by Li et al has *Z*-configuration. *Z*-configuration of the enone is evidently crucial for the formation of DHPIQ.

In agreement with this statement are the experiments between THIQ and cyclohexenone or 3-methylcyclohex-2-en-1-one. None of these compounds gave the respective DHPIQ.

In order to check the effect of the amine structure on the activity in DHPIQ formation, we tested several amines: tetrahydroquinoline, piperidine, dibenzylamine, N-benzylbutan-1-amine. Among these amines, only dibenzylamine and N-benzylbutan-1-amine gave the respective products, although in low yield (determined by GC-MS).

These experiments suggest that activation of the methylene group by the aromatic substituent is important. However, rigid structure of THIQ is significantly more favorable in the condensation reaction with the chalcone compared with a flexible structure of dibenzylamine and N-benzylbutan-1-amine.

This discussion was added to the manuscript, while the article by Chaojun Li etc. in *Org. Lett.*, 2012, 5606 was added to the list of references as a relevant to the present study.

2. Page 4, 108-111: Although THIQ-1,2-d₂ have been used as substrate, a mixture of d-labeled ketones 4a-d₁ is obtained, which will make the reader very confuse to understand the reaction pathway. To this end, THIQ-1,1-d₂ is suggested as an alternative substrate.

Response: We would like to point out that ketones **4a** or **4a-d₁** are formed as side products of chalcone **2a** reduction (ca. 6-10% under the optimized conditions). Ketones **4a** and **4a-d₁** were detected by GC-MS only. The gas chromatograms of the reaction mixtures obtained using THIQ **1a** and THIQ **1a-d₂** and the mass spectra of the corresponding peaks are shown below.

In case of non-labeled experiment between THIQ **1a** + chalcone **2a**, the only signal of ketone **4a** was detected in MS. The distribution of peaks agrees well with the theoretical spectrum due to isotopic abundances of the elements.

In the experiment between chalcone **2a** and labeled THIQ **1a-d₂**, despite the content of *d*-labeled compound was 98%, we still observed as a major product compound with m/z 210.1 that corresponds to ketone **4a**.

From the distribution of m/z intensities in the mass spectrum we conclude that along with the ketone **4a**, a ketone **4a-d₁** bearing one D-atom is also presented in the reaction mixture (increased intensity of the peak with m/z 211.1 compared to the non-labeled experiment).

On the other hand, the intensity of the peak with m/z 212.1 is considerably lower and therefore ketone **4a-d₂** might be present in the reaction mixture but only as a minor component. Summarizing all these observations, a possible kinetic isotope effect may be estimated to be ~2.0. Another possibility for higher yield of ketone **4a** compared to **4a-d₁** might be trivial D-H exchange in ketone **4a-d₁** once it is formed. Such possibility cannot be excluded given that the reaction was performed at +80°C for 20h, at which C-H acidity of acetonitrile becomes high enough to trigger D-H exchange.

Based on the results of the experiment with THIQ **1a-d₂**, we have grounds to conclude that similar experiment with THIQ-1,1-d₂ would also lead to a mixture of isotopically labeled ketones in which position of D-atoms is not clearly defined. Similarly to the abovementioned experiment a large fraction of ketone **4a** free of D-atoms is expected in such experiment. Nevertheless, we agree with the referee that photocatalytic reduction of unsaturated compounds by heterogenous carbon nitride photocatalyst using aliphatic amine as hydrogen source might be an interesting reaction to study. In this case, detailed mechanistic study using different D-labeled compounds without doubts would be necessary.

3. The reaction products may be a class of useful materials, for example, in organic photovoltaics. The author can try to introduce active functional groups (such as bromine, alkyne etc) into chalcone or THIQ, allowing further functionalization of the polycyclic products.

Response: As suggested by the referee, in order to allow for further functionalization of the DHPIQ core, we performed electrophilic bromination of DHPIQ **3d**, as a model compound, using *N*-bromosuccinimide in acetonitrile at room temperature. In this case, bromo-derivative **8d** was isolated in yield 92%.

These results and relevant discussion were added to the manuscript, while ¹H, ¹³C and ¹⁹F NMR spectra of the product **8d** were added to the ESI.

Please note that attempt to introduce bromine atom by using bromo-substituted chalcone as a reaction partner with THIQ under the standard conditions was not successful. As evidenced by GC-MS, a substantial amount of chalcone **2a** formed upon reduction of the bromine-substituted chalcone under the conditions of photocatalysis.

4. The reaction requires three steps: (1) C-N bond formation between THIQ and the carbon atom of the C=O group; (2) C-C bond formation between THIQ and the chalcone; (3) elimination of a H₂ molecule and (4) elimination of a H₂O molecule. For the third step, the author is suggested to detect the generation of hydrogen by GC to identify whether it undergoes elimination of a H₂ molecule.

Response: In order to confirm formation of hydrogen in the photocatalytic reaction of DHPIQ formation, we performed the photocatalytic experiment according to the standard conditions followed by sampling the gas headspace from the reactor with a syringe and injection into the gas chromatograph equipped with thermal conductivity detector (TCD). Indeed, in this experiment we observed a peak that corresponds to H₂. Please note that N₂ peak is apparently due to not perfect connection between syringe barrel and a plunger, because N₂ peak is also observed when Ar was injected into GC-TCD.

This graph and description of the experiment was added to the supporting information of the manuscript.

5. In the page 7, line 174,175, 9p should be changed to 7p. For deuterium-labeling experiment, the deuterium content should be clearly indicated.

Response: We appreciate the referee's very attentive reading the manuscript and pointing to the typos. In the present manuscript d-labeled THIQ (**1a-d₂**) had a content of d-labeled compound 98%. This information was added to the manuscript. The manuscript was revised in order to correct the typos.

Reviewer #3 (Remarks to the Author):

The authors conducted additional experiments to answer the questions in the comments. From my point of view, the questions about the reactions and mechanism have been well answered. The

appropriate background and potential applications of this reaction have also been addressed in the manuscript. With these additional data and edited content, this article is acceptable in the current version.

Response: We appreciate the positive referee's response to accept the manuscript for publication in the current version.

REVIEWERS' COMMENTS:

Reviewer #2 (Remarks to the Author):

The authors have adequately addressed my concerns. I recommend that it could be accepted now.

Replies to the referees' comments to the manuscript "Carbon Nitride Photocatalyzes Regioselective Aminium Radical Addition to the Carbonyl Bond and Yields *N*-fused Pyrroles" by Bogdan Kurpil, Katharina Otte, Artem Mishchenko, Paolo Lamagni, Wojciech Lipinski, Nina Lock, Markus Antonietti and Aleksandr Savateev submitted to Nature Communication.

REVIEWERS' COMMENTS:

Reviewer #2 (Remarks to the Author):

The authors have adequately addressed my concerns. I recommend that it could be accepted now.

Response: We appreciate the positive evaluation of our work and comments provided by the referee earlier.